# The global and multi-annual MUSICA IASI {H$_2$O, $\delta$D} pair dataset

Christopher J. Diekmann[1,a], Matthias Schneider[1], Benjamin Ertl[1,2], Frank Hase[1], Omaira García[3], Farahnaz Khosrawi[1], Eliezer Sepúlveda[3], Peter Knippertz[1], and Peter Braesicke[1]

[1]Institute of Meteorology and Climate Research, Karlsruhe Institute of Technology, Karlsruhe, Germany
[2]Steinbuch Centre for Computing, Karlsruhe Institute of Technology, Karlsruhe, Germany
[3]Izaña Atmospheric Research Center, Agencia Estatal de Meteorología, Santa Cruz de Tenerife, Spain
[a]now at: Telespazio Germany GmbH, Darmstadt, Germany

**Correspondence:** Christopher J. Diekmann (christopher.diekmann@kit.edu)

**Abstract.** We present a global and multi-annual space-borne dataset of tropospheric {H$_2$O, $\delta$D} pairs that is based on radiance measurements from the nadir thermal infrared sensor IASI (Infrared Atmospheric Sounding Interferometer) onboard the Metop satellites of EUMETSAT (European Organisation for the Exploitation of Meteorological Satellites). This dataset is an a posteriori processed extension of the MUSICA (MUlti-platform remote Sensing of Isotopologues for investigating the Cycle of Atmospheric water) IASI full product dataset as presented in Schneider et al. (2021b). From the independently retrieved H$_2$O and $\delta$D proxy states, their a priori settings and constraints, and their error covariances provided by the IASI full product dataset we generate an optimal estimation product for pairs of H$_2$O and $\delta$D. Here, this standard MUSICA method for deriving {H$_2$O, $\delta$D} pairs is extended using an a posteriori reduction of the constraints for improving the retrieval sensitivity at dry conditions. By applying this improved water isotopologue post-processing for all cloud-free MUSICA IASI retrievals, this yields a {H$_2$O, $\delta$D} pair dataset for the whole period from October 2014 to December 2020 with a global coverage twice per day (local morning and evening overpass times). In total, the dataset covers more than 1500 million individually processed observations. The retrievals are most sensitive to variations of {H$_2$O, $\delta$D} pairs within the free troposphere, with up to 30 % of all retrievals containing vertical profile information in the {H$_2$O, $\delta$D} pair product. After applying appropriate quality filters, the largest number of reliable pair data arises for tropical and subtropical summer regions, but also higher latitudes show a considerable amount of reliable data. Exemplary time-series over the Tropical Atlantic and West Africa are chosen to illustrates the potential of the MUSICA IASI {H$_2$O, $\delta$D} pair data for atmospheric moisture pathway studies. Furthermore, in order to facilitate the application of this rather comprehensive MUSICA IASI {H$_2$O, $\delta$D} pair dataset (referred to as Level-2), we additionally provide the data in a re-gridded and simplified format (Level-3) with focus on the quality-filtered {H$_2$O, $\delta$D} pairs in the free troposphere. A technical documentation for guiding the use of both datasets is attached in the appendix. Finally, the Level-2 dataset is referenced with the DOI 10.35097/415 (Diekmann et al., 2021a) and the Level-3 dataset with DOI 10.35097/495 (Diekmann et al., 2021b).

# 1 Introduction

Concomitant observations of moisture content and stable water isotopologues allow fundamental insights into the transport and phase transitions of water in the atmosphere. Differences in the molecular masses lead to characteristic responses of each isotopologue to phase changes. Consequently, the ratio of light and heavy water isotopologues inside an air parcel reveals information about moisture processes that have occurred during its pathway through the atmosphere, and can hence support the investigation of the atmospheric branch of the hydrological cycle (an extensive overview is given in Galewsky et al., 2016). For describing distributions of water isotopologues the $\delta$-notation given in ‰ is commonly used, for instance between $H_2O$ and its heavier isotopologue HDO, with both given as volume mixing ratios:

$$\delta D = \left( \frac{HDO/H_2O}{R_{vsmow}} - 1 \right) \cdot 1000 \tag{1}$$

$R_{vsmow}$ is the isotopic ratio of Vienna Standard Mean Ocean Water as defined by the International Atomic Energy Agency (Craig, 1961). Several studies have proposed the combined analysis of $H_2O$ and $\delta D$ distributions (here denoted as {$H_2O$, $\delta D$} pairs) and demonstrated its value for analysing moisture processes and transport. For instance, signatures in {$H_2O$, $\delta D$} pair distributions from model simulations and measurements were interpreted in terms of the relative contributions of kinetic and equilibrium fractionation, such as Rayleigh condensation, rain evaporation and airmass mixing (e.g., Worden et al., 2007; Noone, 2012; Dyroff et al., 2015; González et al., 2016; Schneider et al., 2017; Eckstein et al., 2018; Lacour et al., 2018; Dahinden et al., 2021; Diekmann et al., 2021c).

During the last decades the space-based remote sensing of tropospheric water isotopologues has progressed considerably in terms of retrieval development, quality and application. On the one hand, cloud-free land observations from short-wave infrared sensors were used to generate total columns of the ratio $HDO/H_2O$ (e.g. Frankenberg et al., 2009; Boesch et al., 2013; Frankenberg et al., 2013; Schneider et al., 2020), while on the other hand thermal infrared sensors allowed for retrieving $HDO/H_2O$ ratios with weak vertical profile information for land as well as ocean observations (e.g. Worden et al., 2006, 2019; Lacour et al., 2012; Schneider and Hase, 2011; Schneider et al., 2016). To ensure coherence in the vertical sensitivities of remotely sensed $H_2O$ and $\delta D$, which is necessary for a combined interpretation, a further post-processing that creates optimal {$H_2O$, $\delta D$} pair information is proposed by Schneider et al. (2012).

However, most of the aforementioned satellite-based water vapour isotopologue retrievals were performed for case studies that were limited in space and time. Up to now, only few global and long-time referenced space-borne datasets of tropospheric $HDO/H_2O$ are available, e.g. total column data from the sensor SCIAMACHY (Scanning Imaging Absorption Spectrometer for Atmospheric Chartography; Schneider et al., 2018) between 2003 to 2012, profile data from TES (Tropospheric Emission Spectrometer; Worden et al., 2012) between 2004 to 2012 (with few available data between 2017 and 2018) and from AIRS (Atmospheric Infrared Sounder; Worden et al., 2019) between 2002 to early 2020. The maximum data availability of these datasets ranges in the order of 1000 to 30,000 observations per day.

In this paper, we present a new global and multi-annual dataset of tropospheric {$H_2O$, $\delta D$} pairs using measurements from the sensor IASI (Infrared Atmospheric Sounding Interferometer). IASI is part of the EUMETSAT Polar System programme (EPS) that comprises the current polar-orbiting satellites Metop-A, Metop-B and Metop-C. The mission started in 2006 and will be

continued from 2022 onwards by its successor EPS-SG (EPS Second Generation), committed to operate the next-generation sensors IASI-NG (IASI Next Generation) onboard of three new Metop satellites for another 20 years. Based on the full swath width of the IASI sensors, this mission is able to provide a global scan of the atmosphere multiple times per day, with about 350,000 cloud-free observations per sensor and per day. The overpasses are designed such that the orbits cross the equator at approximately 9.30 and 21.30 local time (Clerbaux et al., 2009).

To process the enormous amount of IASI measurements, we have set up a quasi-operational processing chain that efficiently runs on high-performance computing clusters (Schneider et al., 2021b). It comprises an extended version of the MUSICA IASI retrieval (Schneider and Hase, 2011). In this context we present the most recent updates regarding the optimal estimation {$H_2O$, $\delta D$} pair product from Schneider et al. (2012), including an a posteriori enhancement of the sensitivity for dry conditions. We discuss and apply a method for achieving an a posteriori reduction of the retrieval constraints. According to the local overpass times, the final results are sorted into morning and evening observations for each day and stored in global output files, respectively. The chosen output format is NetCDF4 and the metadata are in agreement with the conventions for CF (Climate and Forecast) metadata (version 1.7, see https://cfconventions.org/). Additional diagnostic flag variables reflecting the data quality of the retrieved {$H_2O$, $\delta D$} pairs support an intuitive and user-friendly data selection. By post-processing all cloud-free MUSICA IASI retrieval results of Schneider et al. (2021b), we have generated a multi-annual and global dataset of tropospheric {$H_2O$, $\delta D$} pairs for the whole period from October 2014 to December 2020. Accordingly, this dataset is referred to as post-processed Level-2 dataset and, depending on available resources, it is constantly being updated with current IASI measurements.

Since this Level-2 dataset comprises the {$H_2O$, $\delta D$} pair results together with several retrieval metrics (e.g. averaging kernels, uncertainty metrics) for all globally available IASI observations in the given time period, this dataset is rather comprehensive and storage-intensive. For this purpose, we additionally generate a simplified and user-oriented dataset of mid-tropospheric {$H_2O$, $\delta D$} pairs, which only contains the quality-filtered {$H_2O$, $\delta D$} pair data on selected heights of interest and is re-gridded on a regular $1° \times 1°$ grid. This so-called Level-3 dataset has a significantly reduced computational resource requirement and is thus well-suited for larger-scale applications.

The manuscript is organised as follows: Section 2 provides a brief overview of the MUSICA IASI processor and describes details of the improved a posteriori generation of {$H_2O$, $\delta D$} pairs (Level-2). This includes information about the corresponding error treatment and data filtering as well as the generation of a re-gridded Level-3 product. The output volume of the full {$H_2O$, $\delta D$} pair datasets is documented in Section 3. In Section 4, we document the data availability of the full dataset in terms of spatial and temporal coverage. Section 5 shows examples of {$H_2O$, $\delta D$} pair data over the Tropical Atlantic and West Africa for the entire data period. Finally, information about the data access is given in Section 6. A technical user guide for supporting the use of the Level-2 and Level-3 datasets is attached as supplement.

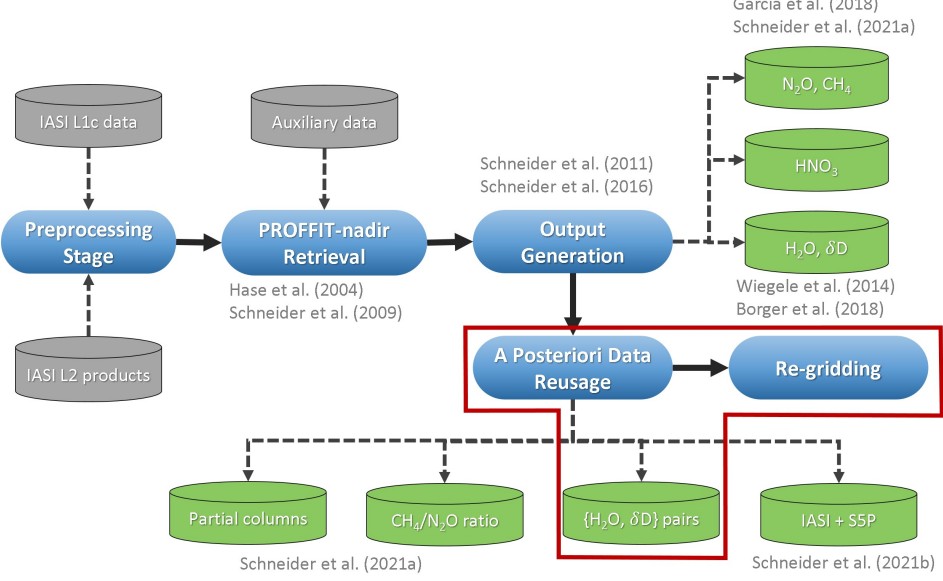

**Figure 1.** Overview of the full MUSICA IASI processing chain and its output products. The red frame indicates the part that is documented in the underlying paper. Further information about this processing chain can be found in Schneider et al. (2021a).

## 2 MUSICA IASI {H$_2$O, $\delta$D} pair post-processing

As part of the MUSICA project (MUlti-platform remote Sensing of Isotopologues for investigating the Cycle of Atmospheric water, 2011 to 2016, Schneider et al., 2016) a retrieval processor was developed and validated for creating {H$_2$O, $\delta$D} pair
information together with N$_2$O, CH$_4$ and HNO$_3$ from IASI spectra. In MUSICA follow-up projects this processing has been further improved and runs now efficiently on high-performance computing clusters. Figure 1 provides an overview of the full processing chain. The main retrieval processing consists of the pre-processing stage, the PROFFIT-nadir retrieval and the output generation and is extensively described in Schneider et al. (2021b). The supply of the full product dataset (Schneider et al., 2021b) offers very good possibilities for data reusage. Examples are an a posteriori synergetic use with products from
other sensors (Schneider et al., 2021a) and the a posteriori generation of an optimal estimation {H$_2$O, $\delta$D} pair product as presented in this paper.

Here, we shortly recall the relevant information of the MUSICA IASI retrieval and subsequently present the improved post-processing for creating and evaluating the {H$_2$O, $\delta$D} pairs.

### 2.1 Main characteristics of IASI

IASI is a Fourier transform spectrometer measuring the thermal infrared upwelling radiation that is affected by atmospheric processes like absorption and scattering. Its spectral resolution is 0.5 cm$^{-1}$. The polar sun-synchronous orbits are designed such that the satellites overpass the equator at 09:30 and 21:30 local time. With around 14–15 orbits per satellite per day and a

full swath width of 2200 km, each IASI sensor achieves a global scan of the atmosphere twice daily. The launches of Metop-A, -B and -C took place in 2006, 2012 and 2018, respectively. The expected lifetime of each satellite is 5 years. Currently, all three Metop satellites are successfully operating in orbit. Further sensor details are listed in Clerbaux et al. (2009).

## 2.2 MUSICA IASI retrieval

The MUSICA IASI retrieval represents an optimal estimation algorithm for retrieving vertical profiles of mixing ratios of water vapour isotopologues and the trace gases $CH_4$, $N_2O$ and $HNO_3$ as well as atmospheric and surface temperatures. It uses the nadir version of the radiative transfer code PROFFIT (Hase et al., 2004) for the spectral window of 1190–1400 cm$^{-1}$ and an iterative Gauss-Newton method for the inversion calculations (Rodgers, 2000; Schneider and Hase, 2011). As proposed in Schneider et al. (2006) and Worden et al. (2006), the retrieval handles the trace gas variations on a logarithmic scale. For water vapour, this enables the use of the retrieval state vectors (ln[$H_2O$] + ln[HDO])/2 and (ln[HDO] - ln[$H_2O$]) that constitute reliable proxies for variations in $H_2O$ and $\delta D$ (Schneider et al., 2012). A concluding post-processing step performs a data compression for large output matrices and creates an output dataset compliant to the CF metadata conventions. For a full technical documentation of the most recent MUSICA IASI retrieval, please refer to Schneider et al. (2021b).

## 2.3 Post-processing for {$H_2O$, $\delta D$} pairs

By considering the MUSICA IASI full retrieval product, we apply a post-processing for optimizing the water isotopologue states and generate an optimal estimation {$H_2O$, $\delta D$} pair product. The following section provides details about the corresponding processing, including information about the error treatment and data selection according to data quality.

### 2.3.1 Vertical representativeness of a remote sensed observation

In general, a retrieved height-depending state vector $\hat{x}$ represents a smoothed image of the true atmospheric state $x_{atm}$ and is defined according to the averaging kernel matrix $\mathbf{A}$ and the a priori state vector $x_a$:

$$\hat{x} = \mathbf{A}(x_{atm} - x_a) + x_a \tag{2}$$

Following the definition by Rodgers (2000), an averaging kernel (rows of $\mathbf{A}$) depicts the fraction of the retrieved result coming from the retrieval itself and not from the a priori assumption. In case of a perfect retrieval the kernel matrix would equal the identity matrix, expressing total independence of the retrieval results from the chosen a priori state. Thus, the degree of deviation from unity quantifies the vertical information content of a remotely sensed observation.

For instance, a common metric for describing the vertical information content is the degree of freedom for signal (DOFS). It is defined as the trace of the averaging kernel matrix. The value of DOFS indicates the number of vertical structures that can be independently determined from an observation (Rodgers, 2000).

Further, the sum of the values along an individual averaging kernel is called measurement response (Eriksson, 2000; Baron et al., 2002). A measurement response of 1 implies that the retrieved state is a smoothed but unbiased image of the true atmospheric profile, whereas values deviating from unity are induced, if the retrieval constraint deviates from pure smoothing

(von Clarmann et al., 2020).

To examine the vertical resolution of a retrieved profile (i.e. the capability to detect vertical structures), metrics that characterize the vertical sensitivity, i.e. the shape of the averaging kernels, can be valuable. First, the relative position of the sensitivity weighted altitude compared to its nominal altitude is termed information displacement. For this, we use the centroid offset as defined by Backus and Gilbert (1970) in a discretized form (Keppens et al., 2015). And second, to describe the vertical smoothing of the retrieved state, the MUSICA IASI retrievals provides two different diagnostics. The definition of Backus and Gilbert (1970) is used to create a kernel-weighted spread around the kernel centroid, while the data density reciprocal of Purser and Huang (1993) serves to indicate the layer width that covers a DOFS of 1. Discussions of these two metrics can be found in Keppens et al. (2015) and von Clarmann et al. (2020), and in the context of the MUSICA IASI full retrieval product in Schneider et al. (2021b).

### 2.3.2 Generation of an optimal estimation {$H_2O$, $\delta D$} pair product

Due to its high variability in the troposphere, $H_2O$ can be detected very well in contrast to $\delta D$, which varies only weakly. As the MUSICA IASI retrieval produces an individual optimal estimation for the different target states, this generally results in different averaging kernels for $H_2O$ and $\delta D$. Therefore, the vertical sensitivity of $H_2O$ is much higher than for $\delta D$ (Schneider et al., 2021b). However, an optimal analysis of {$H_2O$, $\delta D$} pairs would require similar characteristics in the vertical sensitivities to ensure that the retrieved $H_2O$ and $\delta D$ refer to the same vertical structures. Therefore, Schneider et al. (2012) proposed a post-processing for harmonizing the vertical information contents of individually retrieved $H_2O$ and $\delta D$. This is achieved by reducing the strength of the averaging kernels of $H_2O$ with respect to those from $\delta D$. The added value has been proven for IASI results against long-term datasets from ground-based remote sensing stations and in-situ aircraft measurements (Wiegele et al., 2014; Schneider et al., 2015).

As a first step towards harmonizing the sensitivities of $H_2O$ and $\delta D$, we need to transform the water vapour state vector $\hat{x}_{wv}$ and the corresponding averaging kernel block matrix $\mathbf{A}_{wv}$

$$\hat{x}_{wv} = \begin{pmatrix} \hat{x}_{wv,1} \\ \hat{x}_{wv,2} \end{pmatrix}, \ \mathbf{A}_{wv} = \begin{pmatrix} \mathbf{A}_{wv,11} & \mathbf{A}_{wv,12} \\ \mathbf{A}_{wv,21} & \mathbf{A}_{wv,22} \end{pmatrix} \tag{3}$$

from the {$\ln[H_2O]$, $\ln[HDO]$} basis system to the water vapour proxy base {$(\ln[H_2O] + \ln[HDO])/2$, $(\ln[HDO] - \ln[H_2O])$}:

$$\hat{x}'_{wv,a} = \mathbf{P}\hat{x}_{wv,a} \tag{4}$$

$$\hat{x}'_{wv} = \mathbf{P}\hat{x}_{wv} \tag{5}$$

$$\mathbf{A}'_{wv} = \mathbf{P}\mathbf{A}_{wv}\mathbf{P}^{-1} \tag{6}$$

In the following, primed variables are consistently referring to the water vapour proxy state base. Detailed information about the transformation operator $\mathbf{P}$ can be found in Schneider et al. (2012), Wiegele et al. (2014) and Barthlott et al. (2017). $\hat{x}'_{wv}$ is in fact the state that is optimally estimated by the MUSICA IASI retrieval and it represents proxies for $H_2O$ and $\delta D$.

The following step harmonizes the differing sensitivities of the water vapour proxy states by reducing the sensitivity of the

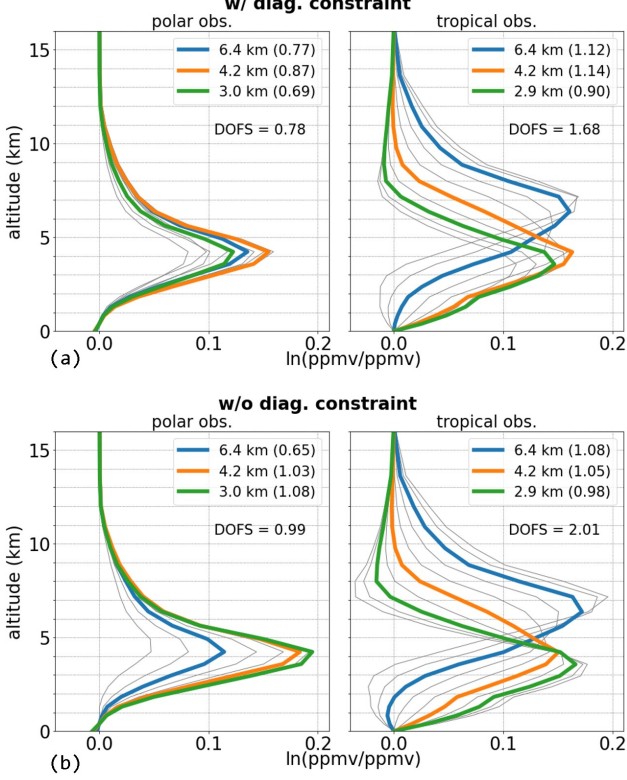

**Figure 2.** Row averaging kernels for the {$H_2O$, $\delta D$} pair product with diagonal constraint (a) and without diagonal constraint (b). The kernels are shown for a polar observation (83.5° N, 147.0° W, left panels in a and b) and for a tropical observation (4.8° N, 45.8° W, right panels in a and b), both above oceans and measured on the 01 July 2017. The measurement response values for the kernels at 3.0, 4.2 and 6.4 km above sea level are given in the respective parentheses in the legends.

$H_2O$ proxy to the sensitivity of the $\delta D$ proxy. The a posteriori correction operator $\mathbf{C}$

$$\mathbf{C}' = \begin{pmatrix} \mathbf{A}'_{wv,22} & \mathbf{0} \\ -\mathbf{A}'_{wv,21} & \mathbf{I} \end{pmatrix} \tag{7}$$

serves to create the harmonized product

$$\mathbf{A}^*_{wv} = \mathbf{C}'\mathbf{A}'_{wv} \tag{8}$$

$$\hat{\boldsymbol{x}}^*_{wv} = \mathbf{C}'(\hat{\boldsymbol{x}}'_{wv} - \hat{\boldsymbol{x}}'_{wv,a}) + \hat{\boldsymbol{x}}'_{wv,a} \tag{9}$$

, which is called Type 2 product in Schneider et al. (2012). The main property of $\hat{\boldsymbol{x}}^*_{wv}$ is that it provides profiles for $H_2O$ and $\delta D$ having practically the same averaging kernels. This allows meaningful analyses of paired {$H_2O$, $\delta D$} distributions.

Figure 2a illustrates the row kernels from $\mathbf{A}^*_{wv}$ for two exemplary {$H_2O$, $\delta D$} pair results above a polar and a tropical site. The main sensitivity lies in the free troposphere, with peaks at $\sim 4$ km above sea level (a.s.l.). The tropical observation shows

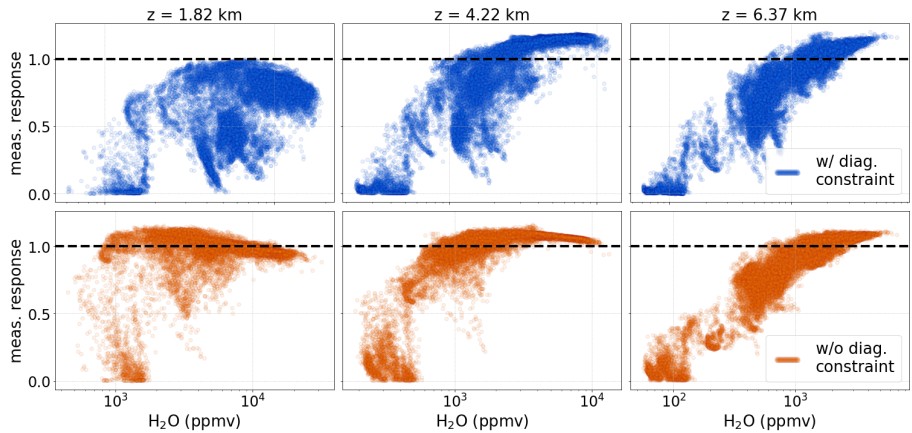

**Figure 3.** Measurement response for the MUSICA IASI {$H_2O$, $\delta D$} pair product along Metop-A orbit 55524 on 01 July 2017, for the original product (blue scatter) and the optimized one (orange scatter). The former includes the diagonal retrieval constraint, while it is removed for the latter. Results are shown for 1.8, 4.2 and 6.4 km above sea level.

additional sensitivity at 6–7 km a.s.l. The kernels for the polar observation show values of measurement response and DOFS falling below 1 by up to 30 % (see values in the parentheses in the legend), meaning there is low vertical sensitivity. In contrast, the {$H_2O$, $\delta D$} pair kernels for the tropical observation exhibit a DOFS of 1.68, though its measurement response still deviates from 1 by about 10–14 %. In the next section we introduce a method that increases the DOFS and the measurement response a posteriori (the corresponding kernels are shown in Fig. 2b, but will be discussed in Sect. 2.3.3).

### 2.3.3 Reduction of retrieval regularization

By inspecting the row kernels in $\mathbf{A}^*_{wv}$ for a full orbit, we observe that there is a general and non-negligible deficit in the measurement response of the {$H_2O$, $\delta D$} pairs. The blue dots in Fig. 3 show that, for instance, at 1.8, 4.2 and 6.4 km a.s.l. and for moisture contents below $10^4$ ppmv a large amount of data contain measurement response values far below 1.

As emphasized in Sect. 2.3.1, the measurement response is a metric for the influence of the a priori assumptions on the retrieval result. Thus, a too low measurement response can be an indicator for a too strong retrieval constraint that excessively pulls the retrieved states towards the a priori profiles (Rodgers, 2000; von Clarmann et al., 2020). Therefore, to reduce the observed lack of sensitivity in the {$H_2O$, $\delta D$} pairs, we apply a method for a posteriori modifying and reducing the underlying retrieval constraint. For this purpose, we adapt a linear optimal estimation method from Rodgers and Connor (2003) that creates a best estimate of a given retrieval result with regards to a new constraint:

$$\mathbf{M}' = \mathbf{R}'^{-1}_d \mathbf{A}'^{T}(\mathbf{A}'\mathbf{R}'^{-1}_d\mathbf{A}'^{T} + \mathbf{S}'_{\hat{\mathbf{x}},\text{noise}})^{-1} \tag{10}$$

The purpose of the operator $\mathbf{M}'$ is that it allows a modification of the retrieval solution $\hat{\mathbf{x}}$ and its kernels $\mathbf{A}'$ according to a weaker constraint. $\mathbf{R}'_d$ is the regularization matrix chosen according to the desired constraint.

In general, a regularization restricts the variability of a retrieved state vector in order to keep the retrieval solution within the

range of physically realistic profiles (Phillips, 1962; Tikhonov, 1963; Rodgers, 2000). By reducing its strength we increase the allowed variability for the retrieved states. As a consequence, the information content increases. On the downside, a weaker constraint causes larger noise and can produce information that is not provided by the measurement (Rodgers, 2000). Therefore, the remainder of this section discusses the optimal definition of the input matrices $\mathbf{R}'_d$, $\mathbf{A}'$ and $\mathbf{S}'_{\hat{\mathbf{x}},\text{noise}}$ as well as the correct usage of $\mathbf{M}'$ in order to enhance the sensitivity of the $\{H_2O, \delta D\}$ pairs.

First of all, to achieve the full benefit from the matrix operations in Eq. (10), we consider the full MUSICA IASI state for the kernel matrix $\mathbf{A}'$ as this is also used during the original retrieval processing. This means that we need to take into account the non-harmonized water vapour proxy state $\mathbf{A}'_{\text{wv}}$ from Eq. (6) and the interfering effects of the other retrieval state vectors. Since the retrieval output from Schneider et al. (2021b) provides only the dominant averaging kernels and cross-correlations, we build $\mathbf{A}'$ as follows:

$$\mathbf{A}' = \begin{pmatrix} \mathbf{A}'_{\text{wv},11} & \mathbf{A}'_{\text{wv},12} & \mathbf{0} & \mathbf{0} & \mathbf{0} & \mathbf{A}'_{\text{t,wv},1} \\ \mathbf{A}'_{\text{wv},21} & \mathbf{A}'_{\text{wv},22} & \mathbf{0} & \mathbf{0} & \mathbf{0} & \mathbf{A}'_{\text{t,wv},2} \\ \mathbf{0} & \mathbf{0} & \mathbf{A}_{\text{ghg},11} & \mathbf{A}_{\text{ghg},12} & \mathbf{0} & \mathbf{A}_{\text{t,ghg},1} \\ \mathbf{0} & \mathbf{0} & \mathbf{A}_{\text{ghg},21} & \mathbf{A}_{\text{ghg},22} & \mathbf{0} & \mathbf{A}_{\text{t,ghg},2} \\ \mathbf{0} & \mathbf{0} & \mathbf{0} & \mathbf{0} & \mathbf{A}_{\text{hno3}} & \mathbf{A}_{\text{t,hno3}} \\ \mathbf{0} & \mathbf{0} & \mathbf{0} & \mathbf{0} & \mathbf{0} & \mathbf{A}_{\text{t}} \end{pmatrix} \tag{11}$$

The kernels of $N_2O$ and $CH_4$ are denoted as $\mathbf{A}_{\text{ghg},11}$ $\mathbf{A}_{\text{ghg},22}$, respectively. The indices 21 and 12 indicate the respective cross-dependencies. The cross-dependency of the temperature retrieval to the water vapour proxy states is marked with $\mathbf{A}'_{\text{t,wv},1}$ and $\mathbf{A}'_{\text{t,wv},2}$. The entries, for which the corresponding kernel matrix are not provided, are filled with the null matrices $\mathbf{0}$.

Further, we calculate $\mathbf{S}'_{\hat{\mathbf{x}},\text{noise}}$ the retrieval noise error covariance, i.e. the variability in the measured radiances that was not explained during the retrieval processing. It can be calculated from $\mathbf{A}'$ and the regularization matrix $\mathbf{R}'$ that was originally applied during the retrieval (Rodgers, 2000; Schneider et al., 2021b):

$$\mathbf{S}'_{\hat{\mathbf{x}},\text{noise}} = \mathbf{A}'(\mathbf{I} - \mathbf{A}')\mathbf{R}'^{-1} \tag{12}$$

Now the question arises about the choice of a new meaningful regularization matrix $\mathbf{R}'_d$. For this purpose, we first recapitulate the original MUSICA approach for setting up the retrieval constraint. For each target species an individual covariance matrix $\mathbf{S}'_a$ is given that describes the potential departure of the retrieval solution from the a priori state. This depends on the choice of the height-depending correlation length, the a priori assumed size of vertical structures that may be resolvable (Schneider et al., 2021b). By inverting $\mathbf{S}'_a$ we yield the corresponding regularization matrix $\mathbf{R}'$. During the MUSICA IASI retrieval the inversion of $\mathbf{S}'_a$ is realized by a decomposition into its diagonal and derivative values (Hase et al., 2004; Schneider et al., 2021b):

$$\begin{aligned} \mathbf{R}' = (\boldsymbol{\alpha}_0 \mathbf{L}_0)^T \boldsymbol{\alpha}_0 \mathbf{L}_0 + (\boldsymbol{\alpha}_1 \mathbf{L}_1)^T \boldsymbol{\alpha}_1 \mathbf{L}_1 \\ + (\boldsymbol{\alpha}_2 \mathbf{L}_2)^T \boldsymbol{\alpha}_2 \mathbf{L}_2 \end{aligned} \tag{13}$$

with $\alpha_i$ as the strength of the individual constraining terms and $\mathbf{L}_i$ as the constraint operators. The diagonal matrices $\alpha_i$ are derived from $\mathbf{S}'_a$ and are provided for each target state as output variables from the MUSICA IASI retrieval (Schneider et al.,

2021b). $\mathbf{L}_0$ represents the diagonal constraint operator and equals the identity matrix $\mathbf{I}$. Its effect is to shift the retrieved profile towards the a priori profile. $\mathbf{L}_1$ and $\mathbf{L}_2$ are the first- and second-order derivative operators and constrain the retrieved profile according to the shape of the a priori covariance, thereby representing smoothing constraints. For the retrieval of atmospheric trace gases with weak spectroscopic signatures smoothing constraints can be advantageous over $\mathbf{L}_0$, because a diagonal con-
straint tightens the retrieval by means of the absolute a priori values, with potentially inducing a bias in the retrieval (e.g. Steck, 2002). Therefore, we infer that the consideration of the diagonal constraint in Eq. (13) causes the observed sensitivity lack in the $\{H_2O, \delta D\}$ pair data for dry conditions. Following this hypothesis, we remove the diagonal constraint operators for the water vapour states and create the new weaker regularization matrix $\mathbf{R}'_{wv,d}$:

$$\mathbf{R}'_{wv,d} = (\boldsymbol{\alpha}_1 \mathbf{L}_1)^T \boldsymbol{\alpha}_1 \mathbf{L}_1 + (\boldsymbol{\alpha}_2 \mathbf{L}_2)^T \boldsymbol{\alpha}_2 \mathbf{L}_2 \tag{14}$$

Keeping the regularization matrices of the other target states unchanged, we can then build the new regularization matrix $\mathbf{R}'_d$ for the full MUSICA state. With that, Eq. (10) is fully determined and we now can use $\mathbf{M}'$ to adjust the kernel matrix $\mathbf{A}'$ according to the new constraint $\mathbf{R}'_d$:

$$\mathbf{A}'_m = \mathbf{M}' \mathbf{A}' \tag{15}$$

Based on the optimized kernel matrix $\mathbf{A}'_m$, we can now create the optimal $\{H_2O, \delta D\}$ pair information for the constraint
reduced state. By extracting $\mathbf{A}'_{wv,m}$ as the first $2\times 2$ block from $\mathbf{A}'_m$, we calculate the new a posteriori operator $\mathbf{C}'_m$ analogous to Eq. (7) and generate the constraint reduced pair product:

$$\mathbf{A}^*_{wv,m} = \mathbf{C}'_m \mathbf{A}'_{wv,m} \tag{16}$$

$$\hat{\boldsymbol{x}}^*_{wv,m} = \mathbf{C}'_m \mathbf{M}' (\hat{\boldsymbol{x}}'_{wv} - \hat{\boldsymbol{x}}'_{wv,a}) + \hat{\boldsymbol{x}}'_{wv,a} \tag{17}$$

This product $\mathbf{A}^*_{wv,m}$ with reduced constraint shows a clear increase of the measurement response (see lower panels in Figure 3).
While the improvements are rather small for 6.4 km a.s.l., the results at 1.8 and 4.2 km a.s.l. have a much better measurement response for moisture contents above 700 ppmv.

The time-series of the measurement response along the orbit used in Fig. 3 is shown in Fig. 4b (upper panel). It is found that the constraint reduction leads to a general decrease of the deviation from 1. Over the Pacific and Atlantic Ocean (observation IDs of 2500–7500 and 15,000–20,000) there is a shift of the slightly over-estimated measurement response towards 1. In
contrast, for higher latitudes its values are originally below 1, but increase significantly due to the constraint reduction. This is in particular pronounced for observations above Australia (observation IDs of 7500–10,000), where an averaged increase in the measurement response of up to 0.5 is apparent. Also for polar observations of the Northern Hemisphere (observation IDS of 0–2500 and 20,000–25,000) the measurement strongly improves.

Analogous improvements become apparent for the individual row kernels in Fig. 2 (compare Fig. 2a and b). The measurement
response increases for the dry polar data at 3.0 and 4.2 km a.s.l. by 56 and 18 %, respectively. Also for the tropical site the measurement response values approach unity. These improvements are not at the expense of vertical resolution, instead they go along with respective improvements in the maximum amplitudes of the individual kernels as well as in the DOFS. For instance,

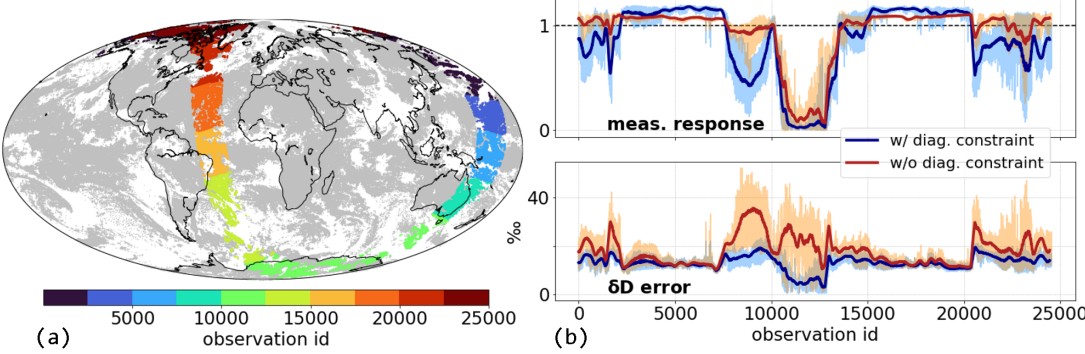

**Figure 4.** Time-series of different retrieval metrics of observations along the Metop-A orbit 55524. (a) shows the location of the observations along the orbit and colour-coded with the chronological observation numbering for this orbit (observation IDs). The gray scatter illustrate all cloud-free Metop-A and -B observations for 01 July 2017 that were considered for the {H$_2$O, $\delta$D} pair post-processing. Panel (b) indicates the time-series of the measurement response for the {H$_2$O, $\delta$D} pairs (upper panel) and the total $\delta$D error (lower panel), both for 4.2 km above sea level. Results for the original product and the product after reducing the retrieval constraint are shown in blue and red, respectively. The solid lines are running means over 200 observations.

the DOFS over the tropical site increases from 1.68 to 2.01, indicating that now information at two different altitude layers can be estimated independently.

### 2.3.4 Error treatment

Several studies have intensively discussed the error treatment for satellite observations in general (Rodgers, 2000; von Clarmann et al., 2020) and with a focus on MUSICA IASI retrieval data (Schneider and Hase, 2011; Borger et al., 2018). Schneider et al. (2021b) provided an overview of the errors that result for the most recent MUSICA IASI retrieval. Along with the kernel modifications for reducing the diagonal constraint for water vapour (see Sect. 2.3.3), a respective processing is required for the dominant MUSICA IASI error covariances.

Given the error covariance $\mathbf{S}'_{\hat{\mathbf{x}}}$ in the proxy state base, we use the optimized a posteriori operator $\mathbf{C}_m$ to transform it according to the reduced constraint:

$$\mathbf{S}'_{\hat{\mathbf{x}},\mathrm{m}} = \mathbf{C}'_{\mathrm{m}} \mathbf{M}' \mathbf{S}'_{\hat{\mathbf{x}}} \mathbf{M}'^T \mathbf{C}'^T_{\mathrm{m}} \tag{18}$$

We perform this processing for the retrieval noise error covariance $\mathbf{S}'_{\hat{\mathbf{x}},\mathrm{noise}}$ from Eq. (12) and for the temperature cross-covariance $\mathbf{S}'_{\hat{\mathbf{x}},\mathrm{temp.}}$:

$$\mathbf{S}'_{\hat{\mathbf{x}},\mathrm{temp.}} = \mathbf{A}'_{\mathrm{t,wv}} \mathbf{S}'_{\mathrm{a,temp.}} \mathbf{A}'^T_{\mathrm{t,wv}} \tag{19}$$

This strongly depends on the choice of the assumed a priori uncertainty covariance $\mathbf{S}'_{\mathrm{a,temp.}}$ (Schneider et al., 2021b).

As these two are the dominant errors for the MUSICA IASI $\delta$D product (Schneider et al., 2021b), we use their sum as an

**Table 1.** Diagnostic flag variables and their recommended values for selecting MUSICA IASI $\{H_2O, \delta D\}$ pair data with high quality. The flags indicating the vertical sensitivity (`musica_wvp_kernel_flag`) and uncertainty (`musica_deltad_error_flag`) of the $\{H_2O, \delta D\}$ pair product are individually set for each altitude level, while the flags for the cloud cover (`eumetsat_cloud_summary_flag`) and the retrieval fit quality (`musica_fit_quality_flag`) are not height-dependent (Schneider et al., 2021a). The vertical sensitivity flag depends on the measurement response, information displacement and vertical resolution of the $\{H_2O, \delta D\}$ pair kernels and the total error flag depends on the sum of the temperature and noise error of $\delta D$.

| Filter type | Variable | Values | Specification |
|---|---|---|---|
| cloud cover | `eumetsat_cloud_summary_flag` | 1 | no cloud contamination |
| | | 2 | minor cloud contaminations possible |
| retrieval fit quality | `musica_fit_quality_flag` | 2 | fair quality |
| | | 3 | good quality |
| vertical sensitivity | `musica_wvp_kernel_flag` | 1 | good vertical representativeness |
| | | | of the averaging kernels |
| total error | `musica_deltad_error_flag` | 1 | $\delta D$ error below 40‰ |

estimate of the total error covariance for the optimized $H_2O$ and $\delta D$ states:

$$\mathbf{S}'_{\hat{\mathbf{x}},\text{tot,m}} = \mathbf{S}'_{\hat{\mathbf{x}},\text{noise,m}} + \mathbf{S}'_{\hat{\mathbf{x}},\text{temp.,m}} \tag{20}$$

The bottom panel in Fig. 4b illustrates how the total $\delta D$ error changes due to the a posteriori constraint reduction. In general, with relaxing the regularization strength the retrieval noise will increase (e.g. Rodgers, 2000). Following this behaviour, the $\delta D$ error exhibits a strong increase for areas where the impact of the regularization optimization is large and the measurement response increases. For instance, the strong improvements of the measurement response over the dry Australian desert are at the

expanse of increasing the averaged $\delta D$ error by 20‰ with single peaks up to 50‰. An increase of the noise is also observed for high latitudes in the Northern hemisphere, whereas for observations above the Pacific and Atlantic Ocean the noise is only slightly affected (compare with discussion in Sect. 2.3.3).

### 2.3.5 Data filtering

Supplementary to the raw IASI L1C measurements, EUMETSAT distributes auxiliary L2 diagnostics, such as cloud cover

and surface type. Utilizing these diagnostics, Schneider et al. (2021b) provide the MUSICA IASI retrieval results for (almost) cloud-free conditions over land, oceans and sea ice, with small cloud contaminations being possible. They supply an additional diagnostic flag variable containing information about the quality of the MUSICA IASI spectral fit. Observations where the simulated spectrum did not converge against the measured spectrum are sorted out from the outset.

As part of the $\{H_2O, \delta D\}$ pair post-processing, we recommend an additional data selection according to the quality of the

retrieved $\{H_2O, \delta D\}$ pair results and share for this purpose further height-dependent flags for an intuitive and user-friendly data handling.

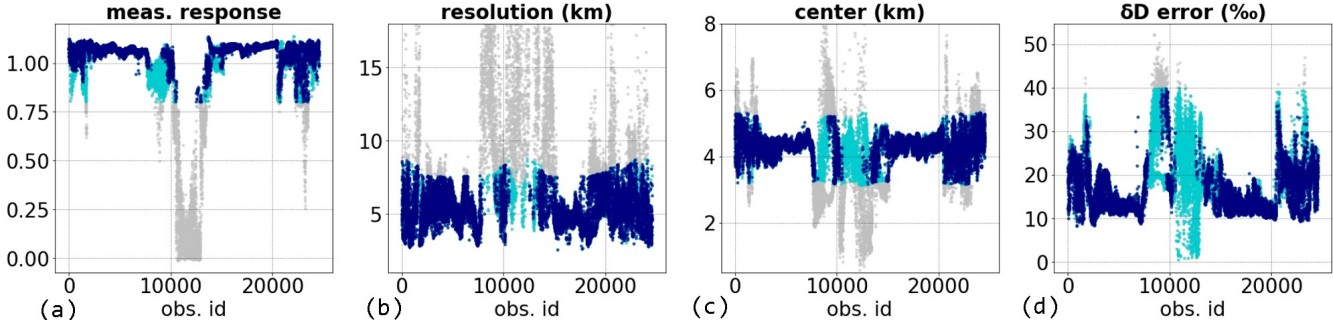

**Figure 5.** Effects of individual filter criteria on the corresponding metrics, along the Metop-A orbit 55524, as also shown in Fig. 4. The grey scatter show all observations along the specific orbit, the cyan scatter show the available data for each variable according to its individual filter criterion and the dark blue scatter are the available data when simultaneously filtering for all four metrics. The metrics (a) – (c) are used to define the flag `musica_wvp_kernel_flag`, and the noise in (d) is used for the flag `musica_deltad_error_flag` (compare with Table 1).

First, we introduce the flag variable `musica_wvp_kernel_flag` for filtering the {$H_2O$, $\delta D$} pairs according to their vertical sensitivity, i.e. to obtain retrieval results that are actually sensitive to the true atmospheric state rather than to the a priori state (see Sect. 2.3.1). Therefore, we define this flag based on the sensitivity metrics of the kernel matrix $\mathbf{A}^*_{wv,m}$. For the measurement response we require values between 0.8 and 1.2. To limit the information displacement at an altitude level $z(i)$, we define following criterion:

$$\frac{|\boldsymbol{c}(i) - z(i)|}{z_{cl}(i)} \leq 0.5 \tag{21}$$

with $\boldsymbol{c}(i)$ being the centroid of the corresponding averaging kernel (Keppens et al., 2015) and $z_{cl}(i)$ the a priori correlation length at the respective altitude level (Schneider et al., 2021b). This criterion ensures that the deviation of the centroid from the nominal height is less than half of the a priori correlation length. As filter condition for the vertical resolution we propose:

$$\frac{\boldsymbol{r}_{LW}(i)}{z_{cl}(i)} \leq 4 \tag{22}$$

$\boldsymbol{r}_{LW}(i)$ is the layer width per one DOFS from Purser and Huang (1993) (see Sect. 2.3.1) as a proxy for the vertical resolution of an averaging kernel. By considering the kernel properties relative to the correlation length, we achieve that also kernels with larger values in their metrics pass the aforementioned filters if larger values in the corresponding correlation length are assumed.

Second, we provide the error flag `musica_deltad_error_flag` that identifies data points with too high uncertainties in the $\delta D$ retrieval results, namely errors due to measurement noise and atmospheric temperature uncertainties. The corresponding height-dependent flag displays retrieval results with a total $\delta D$ error below 40‰.

The aforementioned filter conditions are visualized in Fig. 5 and the respective flags and their recommended values are summarized in Table 1. The flag variables `musica_wvp_kernel_flag` and `musica_deltad_error_flag` are binary,

i.e. they only consist of the values 1 (for indicating high quality) and 0 (for low quality). Even though the recommended filter conditions are chosen somewhat arbitrary, they efficiently remove recognisable outliers in terms of kernel properties (see Fig. 5a–c) and data uncertainties (see Fig. 5d) of the retrieved $\{H_2O, \delta D\}$ pairs. Therefore, the simultaneous application of the corresponding quality flags `musica_wvp_kernel_flag` and `musica_deltad_error_flag` serves for a convenient and meaningful selection of reliable $\{H_2O, \delta D\}$ pair data. However, to enable a flexible adjustment of the individual filter conditions for individual purposes, the output datasets contain the filtered and unfiltered $\{H_2O, \delta D\}$ pair data together with the flag and filter variables.

### 2.3.6 Matrix compression

Analogous to Schneider et al. (2021b), the averaging kernel matrices for the $\{H_2O, \delta D\}$ pairs are stored in a decomposed and compressed format in order to reduce the required storage volume. For this purpose, we apply a singular value decomposition for the matrices $\mathbf{A}^*_{\mathrm{wv},m}$ and $\mathbf{A}'_{\mathrm{t,wv},m}$ into the components $\mathbf{U}$, $\mathbf{D}$ and $\mathbf{V}$ that decompose the kernel matrix through

$$\mathbf{A} = \mathbf{U}\mathbf{D}\mathbf{V}^T \tag{23}$$

The length of the singular value vector $\mathbf{D}$ is called rank. The actual compression is achieved by cutting off the lowest singular values in $\mathbf{D}$ and thereby reducing the rank. Consequently, also the number of singular vectors $\mathbf{U}$ and $\mathbf{V}$ are reduced. The optimal limit of the singular values for balancing the compression error against the effective storage reduction is discussed in Weber (2019). Based on that, we neglect singular values that are less than 0.1 % of the maximum singular value in $\mathbf{D}$.

### 2.4 The final MUSICA IASI $\{H_2O, \delta D\}$ pair product

After performing the aforementioned post-processing and filtering, we obtain the final $\{H_2O, \delta D\}$ pair product, as shown in Fig. 6 for a full Metop orbit. By performing the sensitivity optimization, we observe a substantial increase of variability in the $\{H_2O, \delta D\}$ pairs at 4.2 km a.s.l. for dry regions. For instance, over polar areas the weaker constraints allow larger deviations from the corresponding a priori values, such that lower values in $H_2O$ and $\delta D$ can be observed. This is analogous to the increase in the measurement response that is most pronounced for dry conditions (see Fig. 2, 3 and 4). As the measurement response is considered during the quality filtering for reliable $\{H_2O, \delta D\}$ pairs (see Table 1), its increase yields a higher number of observations passing the recommended data filter (see data amount in Fig. 6).

In summary, the MUSICA IASI $\{H_2O, \delta D\}$ pair post-processing provides an optimal estimation $\{H_2O, \delta D\}$ pair product in the troposphere with a substantial increase of sensitivity for dry conditions. Together with the recommended quality flags indicating observations with meaningful averaging kernels and low errors for $\delta D$, this is the main Level-2 product provided freely to the scientific community.

### 2.5 Generation of a re-gridded $\{H_2O, \delta D\}$ pair product (Level-3 product)

For reasons of traceability and data re-usage, the output files produced by the MUSICA IASI $\{H_2O, \delta D\}$ pair post-processing include arrays for reconstructing different retrieval metrics, such as the averaging kernels and uncertainty covariances. Conse-

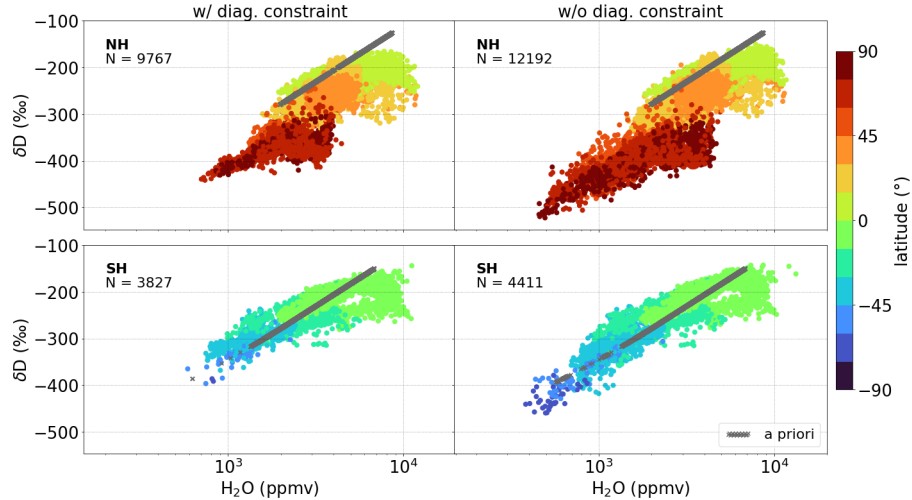

**Figure 6.** Quality filtered {$H_2O$, $\delta D$} pairs (according to Table 1) for the original (with diagonal constraint) and the improved product (without diagonal constraint) at 4.2 km a.s.l. along the Metop-A orbit 55524 during boreal summer (orbit also shown in Fig. 4 and 5). The upper (lower) row shows the scatter for data of the northern (southern) hemisphere, color-coded with the corresponding latitude values. The grey scatter show the a priori values of the individual observations at the nominal altitude. The value of N indicates the respective number of plotted data points.

quently, the corresponding files have high computational requirements with respect to storing and processing. Therefore, we generated an additional Level-3 dataset, the purpose of which is to allow for a simplified and less computationally intensive application of the MUSICA IASI {$H_2O$, $\delta D$} pairs.

As the MUSICA {$H_2O$, $\delta D$} pair product has highest sensitivity typically in the mid-troposphere (between 2.9 and 6.4 km a.s.l.), the Level-3 dataset comprises all quality filtered {$H_2O$, $\delta D$} pairs (according to Table 1) for the fixed altitude levels at 2.9, 4.2 and 6.4 km (analogous to Fig. 2) and re-gridded on a regular $1° \times 1°$ grid. The latter is achieved by linear averaging all data of $H_2O$ and HDO (derived from $\delta D$) within the individual grid boxes and the a posteriori calculation of an averaged $\delta D$ based on the averaged $H_2O$ and HDO data (according to Eqn. (1)). In case of the total errors of $H_2O$ and $\delta D$, the averaging

of their distributions on the regular grid requires to take into account the nature of the errors, i.e. the relative contributions of systematic and random error components. This is crucial, because averaging over systematic error components will balance around a constant systematic bias, whereas the random errors will get smaller the more data points are used for averaging. Here, we follow the simple assumption that errors due to measurement noise and temperature consist of 50% systematic and 50% random error components. We accordingly convolute all measurement noise and temperature error values within the individual

grid boxes and afterwards form the total $H_2O$ and $\delta D$ errors for each grid box. Furthermore, we provide a metric indicating the representativeness of the averaged $H_2O$ and $\delta D$. It is the RMS of the differences of the individual $H_2O$ and $\delta D$ values within the grid boxes to their respective averages. For $H_2O$, the respective calculations are made on the logarithmic scale. As this

metric is a measure for how scattered the individual data are within a single grid box, it indicates the data range for which the averaged value of a single grid box is representative.

## 3   Output volume of the full MUSICA IASI {H$_2$O, $\delta$D} pair datasets

By using the retrieval output from Schneider et al. (2021b), we performed the proposed water isotopologue post-processing for all MUSICA IASI results between October 2014 and December 2020. With on average 350,000 cloud-free observations per sensor (Metop-A and Metop-B) and per day, this results in around 1500 million observations processed for mid-tropospheric {H$_2$O, $\delta$D} pair information.

According to the local overpass times of the Metop satellites, we split the orbits into morning ($\sim$ 09:30 local time) and evening ($\sim$ 21:30 local time) overpasses and concatenate the respective observations for all overpasses within a single day into an individual global NetCDF4 file. That is, two files per day emerge with each having a size of around 4–5 GB, resulting in about 1.7 TB per year. The full output volume is approximately 10.5 TB. This represents the Level-2 dataset of the MUSICA IASI {H$_2$O, $\delta$D} pairs. The Level-3 product described in Sect. 2.5 is generated for all files of the Level-2 dataset, with an approximate individual file size of 4 MB and full output volume of 22 GB. The metadata of the Level-2 and Level-3 output NetCDF4 files are in agreement with the CF metadata conventions. Information about how to access the full dataset is given in Sect. 6. Additionally, we provide a technical documentation for the output files of the Level-2 and Level-3 datasets, which guides the use and application of the MUSICA IASI {H$_2$O, $\delta$D} pair data. This document is provided as supplement to this paper.

## 4   Data coverage and quality

The following section gives an impression of the spatial and temporal representativeness of the optimal estimation {H$_2$O, $\delta$D} pair data. If not otherwise specified, the Level-2 dataset of the MUSICA IASI {H$_2$O, $\delta$D} pairs is used for the analyses and the generation of the figures throughout the following section.

### 4.1   Degree of freedom for signal

Figure 7 shows the DOFS values of $\delta$D as monthly means for February and August 2018, for morning and evening observations, respectively. To consider the full quality range of the {H$_2$O, $\delta$D} pair results, we here investigate the $\delta$D distributions filtered only for cloud-free scenes and acceptable retrieval fit quality, but not for the {H$_2$O, $\delta$D} pair quality (only filtered for `musica_fit_quality_flag` $\geq 2$ and `eumetsat_cloud_summary_flag` $\leq 2$).

Maximum values are around 2 and are found over the tropics (persistently throughout the entire year) and the sub-tropics (during summer), indicating the capability of independently resolving signals in the lower and middle free troposphere (as indicated by the averaging kernels in Fig. 2). The DOFS minimum is located over the polar regions during winter times, as these regions are typically very dry and cold.

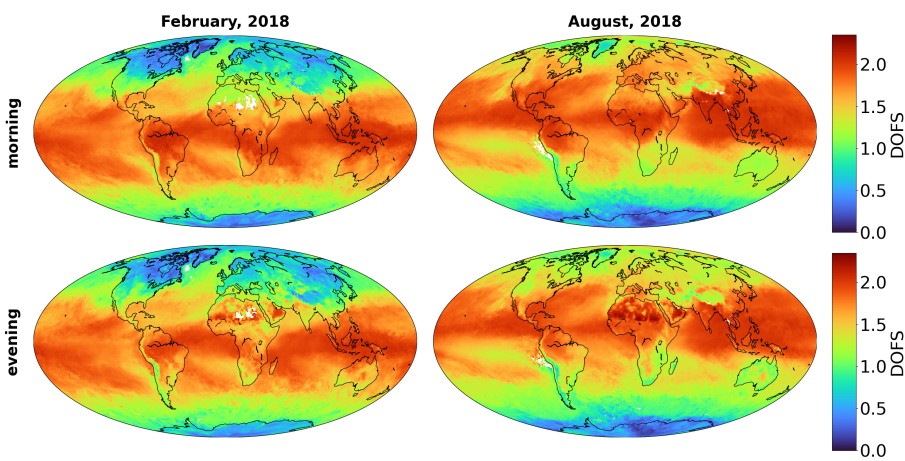

**Figure 7.** Monthly averages for February and August 2018 for the DOFS of the {$H_2O$, $\delta D$} pair product without {$H_2O$, $\delta D$} pair quality filtering, evaluated on a $1° \times 1°$ grid.

Over oceans, the DOFS distribution roughly reflects the dominant sea-surface temperature patterns. For instance, the warm surface currents in the West Atlantic and West Pacific correlate with an increased sensitivity of the {$H_2O$, $\delta D$} pair retrievals. While the large-scale DOFS patterns show a strong inter-annual variability for all regions except the tropics, their diurnal variations are rather small. Instead, the latter becomes more pronounced for small-scale regional structures. In particular for land observations thermal effects lead to a sensitivity maximum for morning times (Clerbaux et al., 2009), e.g. for Australia during February and for Europe and North America during August. Conversely, for the Sahara we observe an inverted effect, i.e. an increase of DOFS from morning to evening. As a next step, we will consider data that have been additionally filtered for high sensitivity and low uncertainty in the {$H_2O$, $\delta D$} pair product (see Table 1).

## 4.2 Vertical distribution of data coverage

As discussed in Sect. 2.3, the MUSICA IASI water vapour retrieval is mainly sensitive to water vapour in the free troposphere. Figure 8 shows that this is reflected clearly on the vertical distribution of available {$H_2O$, $\delta D$} pairs after applying the full recommended filters according to Table 1. Here, the amount of globally available observations per day and per morning and evening maps is averaged for February and August 2018 and is shown for each retrieval grid level between the surface and 9 km. The best data availability arises between 2–7 km a.s.l. On average, during boreal summer (at maximum over 400,000 data pairs per day) remarkably more observations are available than during austral summer (maximum 300,000 data pairs). In contrast, the diurnal variations are again rather small on the global scale. Only for altitudes below 3.5 km a.s.l. we observe a slight decrease of data availability during evening. This might be due to thermal heating that develops during the day and leads to a upwards transport of low-level moisture, resulting in a upwards shift of the retrieval sensitivity. Such effects are stronger over land masses and during summer and probably lead to a larger morning-to-evening difference during boreal summer, as

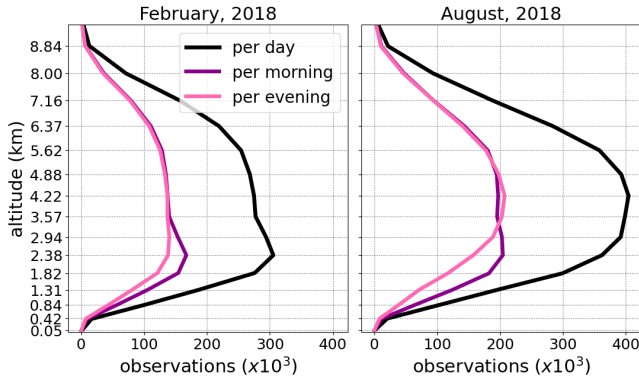

**Figure 8.** Averaged amount of quality filtered {$H_2O$, $\delta D$} pairs for the tropospheric retrieval grid altitudes during February and August 2018. The black line indicates the global means that are further divided into the means for morning (violet) and evening (pink) overpasses.

there are more land masses in the Northern than in the Southern Hemisphere.

## 4.3 Horizontal distribution of data coverage

In this section we discuss the horizontal data coverage of {$H_2O$, $\delta D$} pairs for different altitude regions after applying the respective quality filtering according to Table 1. To identify those retrieval results that provide vertical profile information, we check for observations that fulfil the respective filter criteria simultaneously for two distinct altitudes.

Figure 9 shows the mean horizontal coverage of quality-filtered {$H_2O$, $\delta D$} pairs for different altitude regions during February and August 2018; the corresponding IASI observations are evaluated on a $1° \times 1°$ grid. The averaged number of daily available

{$H_2O$, $\delta D$} pairs and the fraction of days with at least one measurement are illustrated for each grid box. Additionally, Table 2 provides the total fractions of available {$H_2O$, $\delta D$} pairs on each altitude region compared to all cloud-free IASI observations. At 4.2 km a.s.l., up to 59 % of all cloud-free IASI observations provide reliable {$H_2O$, $\delta D$} pair data, with best horizontal coverage over tropical and subtropical summer regions. Here, up to 35 observations are available per day and grid box and over wide areas there is a 100 % frequency of $1° \times 1°$ grid boxes with at least one reliable observation, especially in the

tropics and the summertime sub-tropics. But also for high northern latitudes, where typically cold and dry conditions prevail, a satisfactory data availability is apparent. Furthermore, for about 22–30 % of the cloud-free observations the quality filter conditions are simultaneously fulfilled at 2.9 and 6.4 km a.s.l. We observe similar spatial patterns with lower values and less temporal coverage, when compared to 4.2 km a.s.l. Even though the data coverage decreases significantly for areas with profile information at even lower altitudes (the quality filter conditions are simultaneously fulfilled at 1–1.5 and 4–5 km above ground

level only for about 10–17 % of the cloud-free observations), interesting features emerge. The maximum availability of about 10 observations per grid box and per day shifts towards higher latitudes, such that over the tropics there are almost no data.

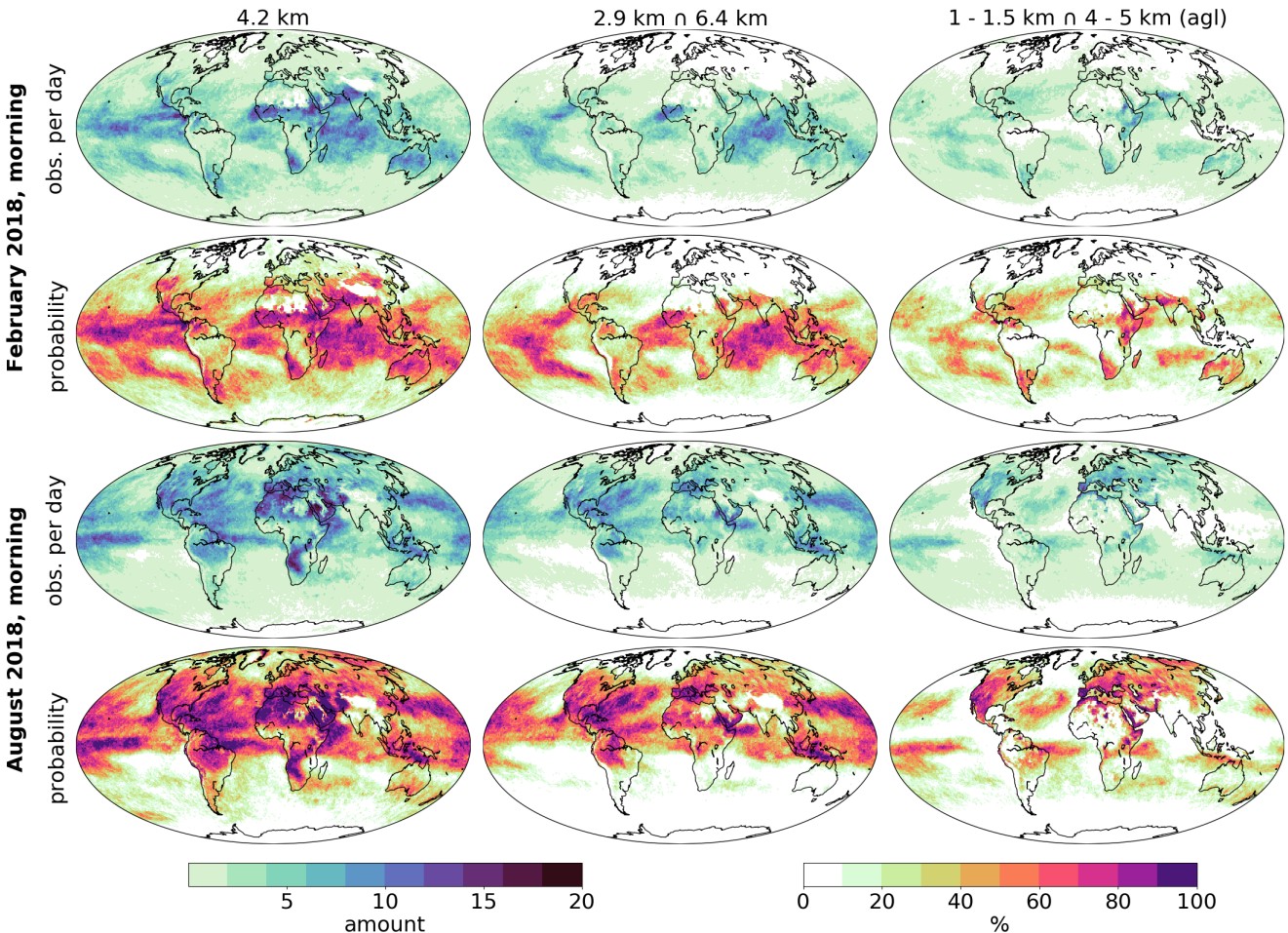

**Figure 9.** Monthly statistics for the horizontal availability of MUSICA IASI {$H_2O$, $\delta D$} pair data for February 2018 (first and second row) and for August 2018 (third and forth row). Data are filtered according to Table 1. The first row for each month, respectively, shows the averaged amount of available observations per $1°\times1°$ grid box and per day, the respective second row gives the frequency of days with at least 1 reliable observation inside a single $1°\times1°$ grid box. Shown are the results for observations at 4.2 km above sea level (a.s.l.) in the first column, for observations fulfilling the quality filter conditions simultaneously at 2.9 and 6.4 km a.s.l. (second column) and simultaneously at 1–1.5 and 4–5 km above ground level (a.g.l.; third column). For the latter, if more than one grid level falls inside the given altitude range, then the lower one is chosen.

In this analysis we jointly investigated the morning and evening observations. As can be deduced from Fig. 7 and 8, the differences between the morning and evening distributions will differ only little. For instance, Table 2 includes the fractions of available data after filtering according to Table 1 for the altitude regions from Fig. 9. The values do not differ significantly for the mid-troposphere during morning and evening times, but reduce for lower altitudes during the evening overpasses (analogous to Fig. 8).

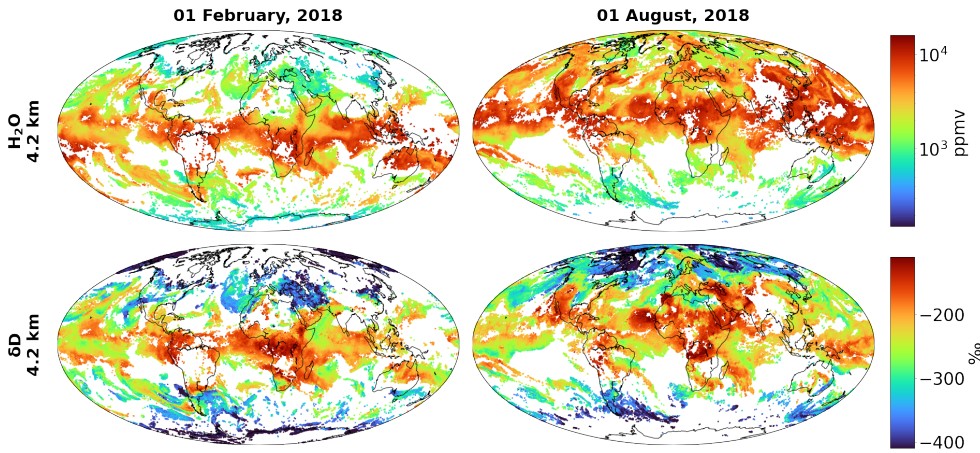

**Figure 10.** Horizontal distributions of $H_2O$ and $\delta D$ from the optimal estimation pair product, after filtering according to Table 1. Data are shown for 4.2 km a.s.l. and for 01 February and 01 August 2018 (including both morning and evening observations). The range of the colorbars is adjusted to Fig. 15 of Schneider et al. (2021b).

**Table 2.** Averaged fractions of available MUSICA IASI $\{H_2O, \delta D\}$ pair data after applying the quality filter according to Table 1, compared to the full (i.e. unfiltered) cloud-free IASI observations. The results are shown for 4.2 km above sea level (a.s.l.), for observations where the filter conditions are fulfilled simultaneously at 2.9 and 6.4 km a.s.l. and at 1–1.5 and 4–5 km above ground level (a.g.l.), respectively. For the latter, if more than one grid level falls inside the given altitude range, then the lower one is chosen.

| Date | Overpass | 4.2 km (a.s.l.) | 2.9 km ∩ 6.4 km (a.s.l.) | 1–1.5 km ∩ 4–5 km (a.g.l.) |
|------|----------|-----------------|--------------------------|----------------------------|
| Feb. 2018 | morning | 41.4 % | 21.7 % | 13.0 % |
|           | evening | 40.7 % | 20.1 % | 9.8 % |
| Aug. 2018 | morning | 57.5 % | 30.1 % | 23.0 % |
|           | evening | 58.8 % | 27.1 % | 11.2 % |

To convey an impression of the actual horizontal data distribution of the $\{H_2O, \delta D\}$ pair product, Figure 10 depicts all data of $H_2O$ and $\delta D$ at 4.2 km a.s.l. for two days (01 February and 01 August 2018). The horizontal patterns of available data are in agreement with Fig. 9. Both $H_2O$ and $\delta D$ show highest values over tropical regions and decrease towards the polar areas. However, differences in their zonal distribution become apparent. For instance, while $H_2O$ and $\delta D$ show consistently high values over Northern Africa, large discrepancies appear at similar latitudes over the Pacific (high $H_2O$ combined with decreased $\delta D$). Section 5 will give further insights into such relations between $H_2O$ and $\delta D$.

### 4.4 Horizontal distribution of data uncertainty

Figure 11 shows the horizontal distributions of the total errors of $H_2O$ and $\delta D$ at 4.2 km a.s.l., exemplarily for 01 February and 01 August 2018. Overall, an anti-correlation to the DOFS distributions in Fig. 7 may be identified. The lowest errors are

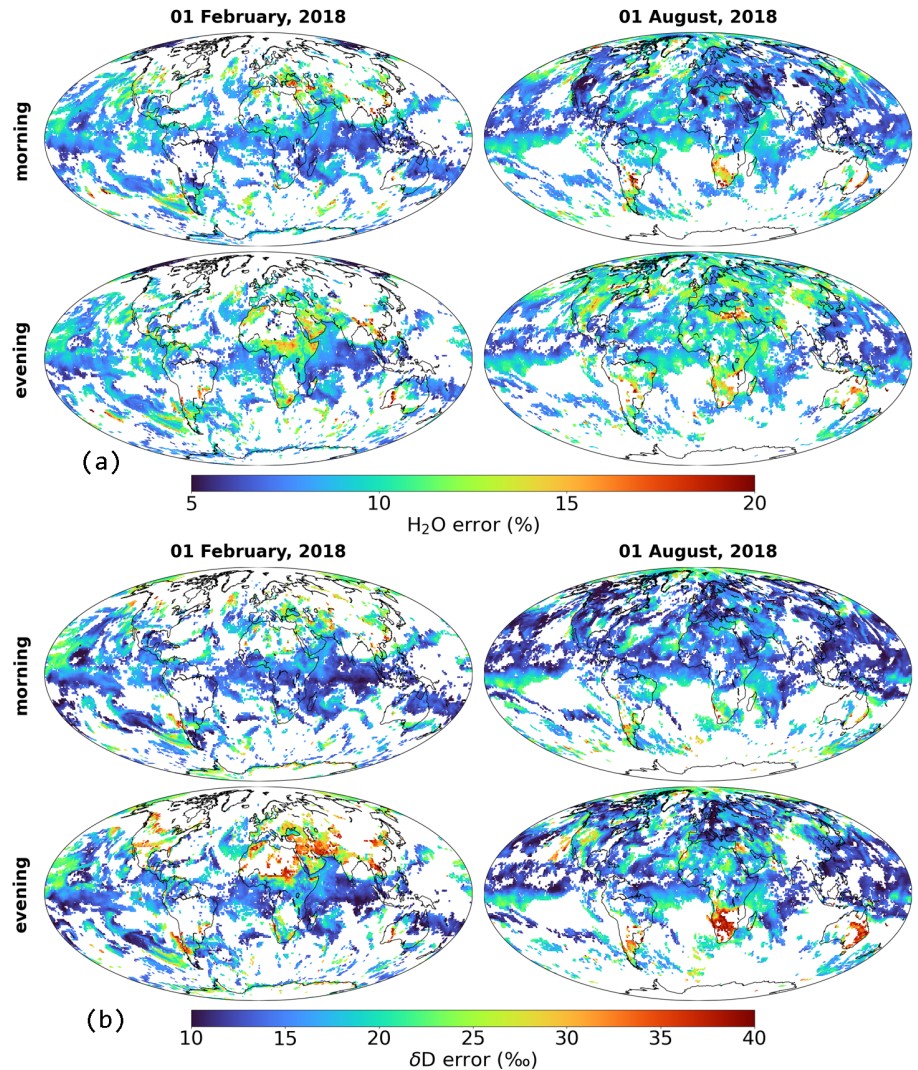

**Figure 11.** Distributions of the total errors of the filtered $H_2O$ (a) and $\delta D$ (b) product at 4.2 km a.s.l., shown for the morning and evening data of 01 February and 01 August 2018. The filtering is performed according to Table 1.

found for warm and moist tropical and sub-tropical sites during summer, where the DOFS is maximum. Here, the minimum error values lie around 5 % and 10 ‰ for $H_2O$ and $\delta D$, respectively. With decreased sensitivity during winter and for higher latitudes, we observe an increase of the total errors, in particular over land areas. The errors can reach values up to $\sim 12$ % and 30 ‰ for $H_2O$ and $\delta D$, but are still in the range of uncertainty from other comparable remotely sensed products (Worden et al., 2006, 2019). In contrast to these Level-2 data errors, the averaged $H_2O$ and $\delta D$ errors of the Level-3 data product will be overall lower (not shown), which is a result of the averaging over the assumed random error components (see Sect. 2.5). Furthermore, Fig. 12 provides an overview of the representativeness of the $H_2O$ and $\delta D$ data being averaged for the regular

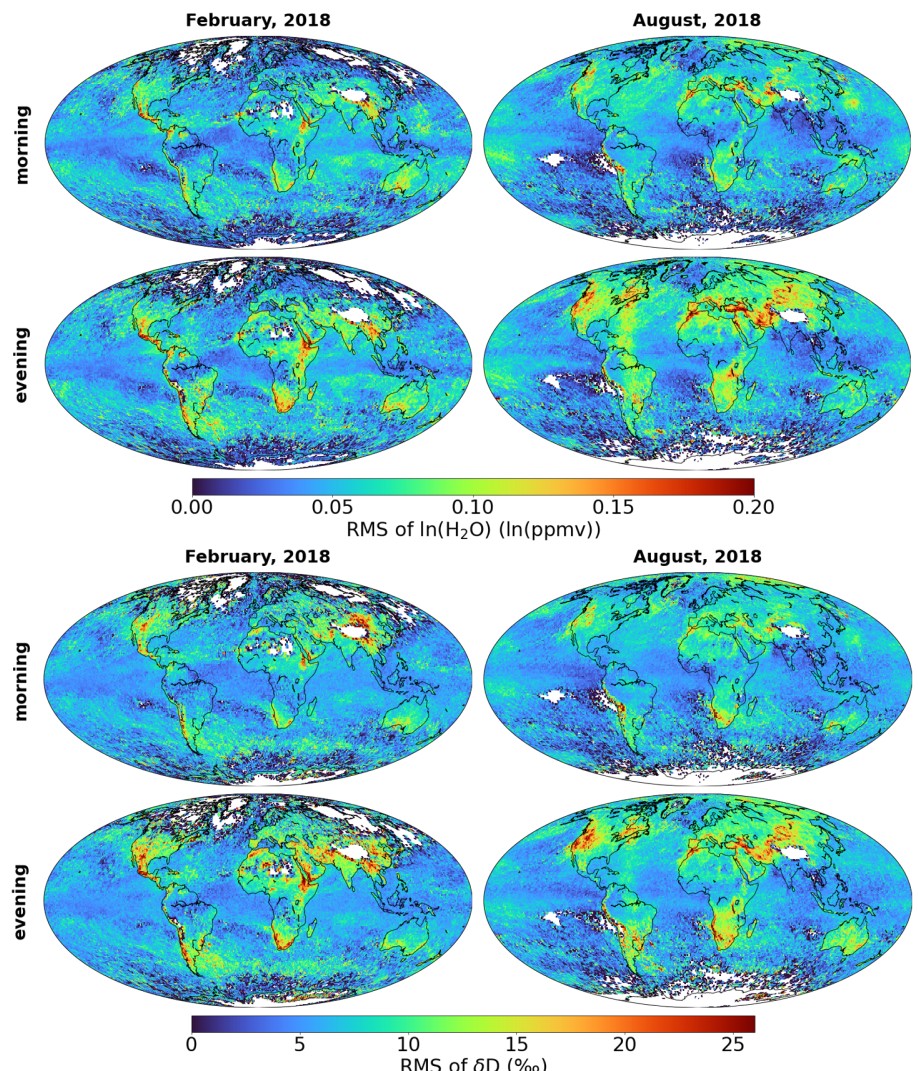

**Figure 12.** Monthly averages of the RMS difference of all data points within the individual $1° \times 1°$ grid boxes to their averaged value, evaluated at $4.2\,\mathrm{km}$ a.s.l. for the quality-filtered $H_2O$ data in logarithmic scale (a) and the quality-filtered $\delta D$ data (b). The filtering is performed according to Table 1.

$1° \times 1°$ grid, as done for the Level-3 dataset of the MUSICA IASI $\{H_2O, \delta D\}$ pairs (see Sect. 2.5). For this purpose, monthly means of the representativeness metrics (RMS values, see Sect. 2.5) are shown for Februar and August 2018. The lowest
RMS values appear for both $H_2O$ and $\delta D$ over oceans, meaning that these regions exhibit rather homogeneous and compact distributions in $H_2O$ and $\delta D$ within the individual grid boxes. In contrast, the highest RMS values arise for coastal regions in the subtropics, where due to the land-sea-contrast the largest spread of $H_2O$ and $\delta D$ values within the individual grid boxes develops.

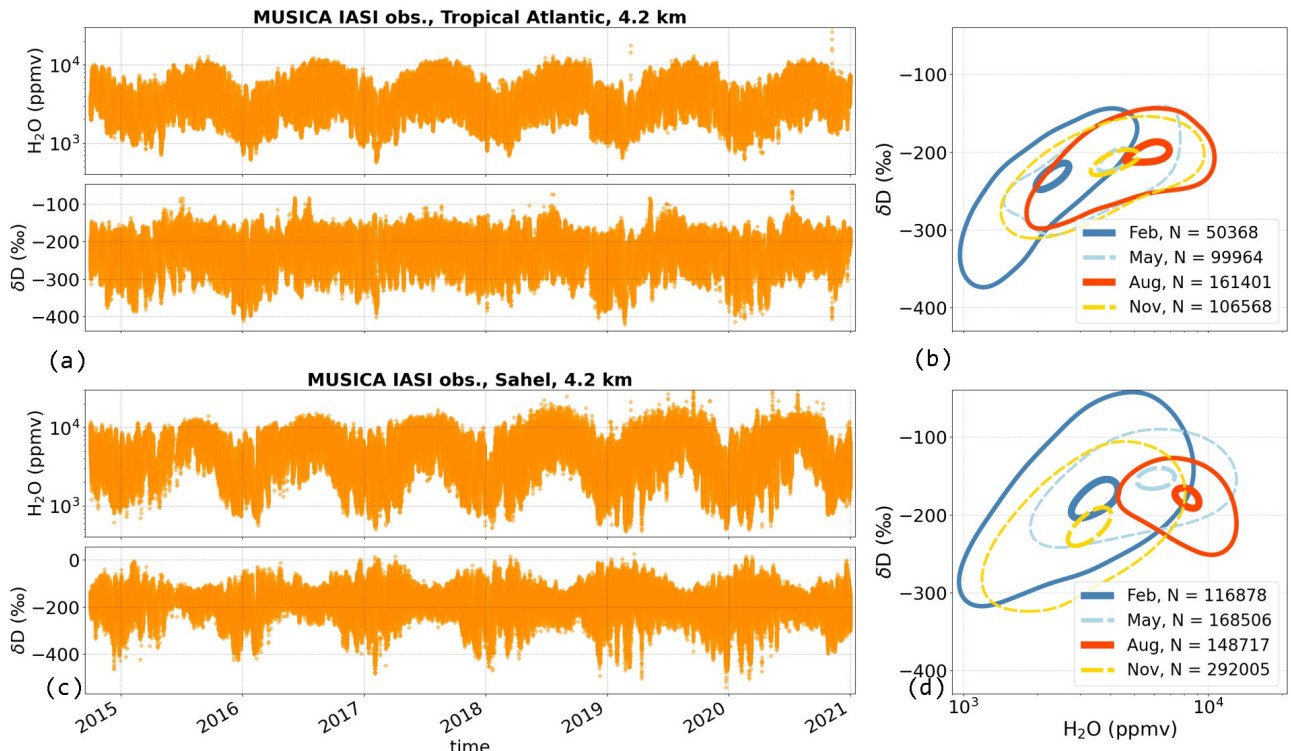

**Figure 13.** MUSICA IASI {$H_2O$, $\delta D$} pair data for 4.2 km a.s.l. over the Tropical Atlantic (13–17° N, 46–30° W) and the Sahel in West Africa (13–17° N, 8° W – 8° E), for the full MUSICA IASI period. The left plots show the time series of $H_2O$ and $\delta D$. The right plots show the respective probability density functions of the two-dimensional {$H_2O$, $\delta D$} distributions, indicating the location of the main 10 and 90 % scatter points. These contours are drawn for the data of February, May, August und November, summarized for all respective years.

## 5  Data example: Tropical Atlantic and Sahel

To convey an impression of the amount and scientific potential of the optimal estimation MUSICA IASI {$H_2O$, $\delta D$} pair product, we discuss results for two illustrative regions of interest, namely the Tropical Atlantic (13–17° N, 46–30° W) and the Sahel in West Africa (13–17° N, 8° W – 8° E).

Figure 13 shows the time-series of the respective MUSICA IASI data for $H_2O$ and $\delta D$ at 4.2 km a.s.l. that have passed the full recommended filtering (according to Table 1) for the period October 2014 to December 2020. As discussed in Section 2.3.2,
the harmonized retrieval results for $H_2O$ and $\delta D$ offer almost the same averaging kernels, thereby allowing for a meaningful interpretation of paired {$H_2O$, $\delta D$} distributions. Based on that, Figure 13 also summarizes the mean monthly evolution (represented by February, May, August and November) of the {$H_2O$, $\delta D$} pair distribution over the Tropical Atlantic and the Sahel. The data are illustrated with normalized two-dimensional histogram contours comprising the main 10 and 90 % of the scatter points (the calculation is described in the appendix of Eckstein et al. (2018)).

Over the Tropical Atlantic, both $H_2O$ and $\delta D$ exhibit a similar annual cycle, even though it is weaker for $\delta D$. This can also be

observed in the corresponding {$H_2O$, $\delta D$} pairs, where the August data are on average moister and more enriched in $\delta D$ than the February data. Despite some shifting and tilting, the overall contour shape does not change to first order from February to August.

In contrast, over the Sahel signs of an annual anti-correlation between $H_2O$ and $\delta D$ appear. Again, during February there is a minimum of $H_2O$ and $\delta D$, even though it is slightly moister than over the Tropical Atlantic. During boreal summer, the variability of $H_2O$ and $\delta D$ decreases significantly, while the respective contours shift to higher $H_2O$. However, this moistening is associated with a strong decrease of the maximum values in $\delta D$, leading to a remarkable tilting of the August contour, when compared to the February contour.

These regional differences highlight the benefit of adding information about $\delta D$ when studying atmospheric moisture, because different moisture processes leave different impact on the shape and position of {$H_2O$, $\delta D$} pair distributions. In the example of Fig. 13, we observe that the Tropical Atlantic and the Sahel reveal significantly different structures in $\delta D$, while their $H_2O$ distributions show clear and similar annual cycles. Therefore, this feature makes clear that the tropospheric moisture over the two tropical regions is governed by structurally different processes. As $\delta D$ is mainly affected during phase changes of water vapour, we infer that its observed anti-correlation to $H_2O$ may be an effect of tropical convection that is exceptionally strong over the Sahel during the summertime monsoon period. Related dynamical changes in the contributing wind regimes might pose further contributing factors for changes in the {$H_2O$, $\delta D$} phase space.

However, in order to robustly attribute such {$H_2O$, $\delta D$} pair signals to underlying moisture processes, supplementary measurements and model analyses are required. As previous studies stated (e.g. Worden et al., 2007; Noone, 2012; Dyroff et al., 2015; González et al., 2016; Schneider et al., 2017; Christner et al., 2018; Eckstein et al., 2018; Lacour et al., 2018; Dahinden et al., 2021; Diekmann et al., 2021c), such an analysis is then capable of providing a deeper understanding of atmospheric moisture pathways and will therefore be part of future MUSICA IASI studies.

## 6 Dataset availability

The Level-2 dataset of the MUSICA IASI {$H_2O$, $\delta D$} pair product is referenced with the DOI 10.35097/415 (Diekmann et al., 2021a). In its description, this DOI refers to the data available from October 2014 to June 2019, because only this period was available at the time of the DOI assignment. However, this dataset could be extended to additionally include all data until December 2020. The Level-3 dataset of the MUSICA IASI {$H_2O$, $\delta D$} pairs is referenced with the DOI 10.35097/495 (Diekmann et al., 2021b). The full Level-2 and Level-3 datasets are freely available via the web portal http://www.imk-asf.kit.edu/english/musica-data.php.

## 7 Summary

We present an extension of the MUSICA IASI retrieval that aims at creating an optimized water isotopologue pair product for the free troposphere. The retrieval processor from Schneider et al. (2021b) is an update of the version that was developed and

validated against reference measurements during the MUSICA project (Schneider et al., 2016). The presented post-processing step exploits their retrieval results and generates an optimal estimation {$H_2O$, $\delta D$} pair product by harmonizing the averaging kernels of $H_2O$ and $\delta D$, as proposed by Schneider et al. (2012). We introduce a further optimization step by a posteriori reducing the strength of the underlying regularization. This increases the sensitivity of the {$H_2O$, $\delta D$} pair retrieval product, especially for dry conditions, and enhances the vertical profile information between the boundary layer and the free troposphere. However, as trade-off the retrieval noise increases, but not beyond an unreasonable range ($\sim 12\,\%$ for $H_2O$ and $\sim 30\,‰$ for $\delta D$). For a user-friendly data handling, we derive supplementary filter flags that perform a height-depending data selection based on the quality of the {$H_2O$, $\delta D$} pair results. An additional technical user guide attached as supplement aims to support and facilitate to work with the {$H_2O$, $\delta D$} pair data.

We applied this post-processing to the MUSICA IASI full retrieval product and created a novel space-borne dataset of tropospheric {$H_2O$, $\delta D$} pair data. It consists of two global maps per day for all cloud-free IASI observations between October 2014 and December 2020. On a global average, the main vertical sensitivity lies between 2–7 km. It features best horizontal representativeness in terms of data quality and coverage for tropical and summertime sub-tropical regions. Despite a negative equator-to-pole gradient in the horizontal representativeness, there is still a satisfactory amount of reliable {$H_2O$, $\delta D$} pair data in higher latitudes, with ranging during summer up to polar regions. In addition to this comprehensive Level-2 dataset of MUSICA IASI {$H_2O$, $\delta D$} pairs, also a reduced Level-3 dataset is provided, which consists of mid-tropospheric {$H_2O$, $\delta D$} pairs re-gridded on a regular $1° \times 1°$ grid.

Due to the unprecedented combination of high coverage and resolution in space and time, the MUSICA IASI {$H_2O$, $\delta D$} pair datasets are highly promising for studying atmospheric moisture pathways. It enables analyses across different scales, from annually to daily, from globally to locally, and are therefore appealing to a wide range of scientific applications. For further encouraging the use of these data, the full Level-2 dataset is made freely available to the scientific community under the DOI 10.35097/415 (Diekmann et al., 2021a) and the Level-3 dataset unter the DOI 10.35097/495 (Diekmann et al., 2021b).

*Author contributions.* FH developed the radiative transfer model PROFFIT-NADIR. BE and MS optimized the MUSICA IASI retrieval. BE, MS, ES and OG performed the retrieval calculations. MS, CD and FK developed the water isotopologue post-processing. CD performed the water isotopologue post-processing and created the data statistics. CD wrote major parts of the manuscript. PB and PK supervised the PhD of CD. All authors contributed to the discussion of the paper.

*Competing interests.* The authors declare that they have no conflict of interest.

*Acknowledgements.* This work has strongly benefited from the project MUSICA (funded by the European Research Council under the European Communitys Seventh Framework Programme (FP7/2007-2013)/ERC Grant Agreement number 256961), from financial support in the context of the projects MOTIV and TEDDY (funded by the Deutsche Forschungsgemeinschaft under project IDs/Geschäftszeichen

950290612604/GZ:SCHN1126/2-1 and 416767181/GZ:SCHN1126/5-1, respectively) and INMENSE (funded by the Ministerio de Economía y Competividad from Spain, CGL2016-80688-P).

The here published dataset was generated using the supercomputer ForHLR, which is funded by the Ministry of Science, Research and the

Arts Baden-Württemberg and by the German Federal Ministry of Education and Research. Further, the authors wish to acknowledge the contribution of Teide High-Performance Computing facilities. TeideHPC facilities are provided by the Instituto Tecnolo y de Energ Renovables (ITER), S.A (teidehpc.iter.es).

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
