# Peer review of "The global and multi-annual MUSICA IASI $\{\text{H}_2\text{O}, \delta\text{D}\}$ pair dataset"

_Earth System Science Data, 2021_

## Referee Comment (RC1)

**Review of Diekmann et al, ESSD**

April 16, 2021

**1   General comments**

This paper presents a re-processed dataset of paired water vapor and $\delta D$ retrievals from satellite observations with the IASI instrument. This is very useful dataset for the community of people working on water vapor isotopes. To date, this dataset will become the isotopic dataset with the highest spatio-temporal sampling (twice daily) at the global scale with the longer time period (six years). This submission is thus well-suited for publication in ESS.

The article is well-written and well-illustrated.

I expect that many users will be interested in using this data. However, the complexity of the dataset and of its use may be dissuasive. I propose a few suggestions in the specific comments.

Note that I'm just a data-user. I'm not expert in the satellite data processing, so I cannot comment on this aspect. I hope that at least another reviewer will be such an expert.

**2   Specific comments**

**2.1   Optional: what about adding a user-friendly, fully gridded product?**

The full dataset is extremely heavy. IASI is not the first global dataset of water vapor isotopic composition by satellite, and my impression is that previous such datasets have been under-exploited. I can feel that many users would like to use such datasets, but the size of the data, and the complexity of its processing by users, is dissuasive for many people.

Therefore, I think it would be very useful for the community to build a simpler, lighter, regularly-gridded, user-friendly product. For example, it could be one netcdf file for each month, with a daily resolution, a 1x1 resolution, and with a coarse, horizontally-uniform vertical grid in altitude or pressure. It would contain quality filtered humidity and $\delta D$ profiles and just a few ancillary information. Maybe you already made such a product, for example when plotting some gridded maps for this article?

Have you ever considered making such a product publicly available?

**2.2   Clarify how the data should be used**

I downloaded the dataset on the link given line 436 and tried to use it myself to test if the article was clear enough for users. I had several difficulties.

- Table 1: it is indicated that the flag *eumetsat_cloud_summary_flag* should be 1 or 2. However, l 337, the condition is written as *eumetsat_cloud_summary_flag*$\leq$2. So should we select or discard values of *eumetsat_cloud_summary_flag* that are 0?

- I don't understand what is the *musica_species_id* dimension in the data files. For example, how do you plot figure 7 using the variable *musica_wvp_dofs*?

- If I want to compare model outputs to the IASI data, is there a simple formula that would allow me to convolve model outputs with the IASI averaging kernels? I expect that I would have to use *musica_h2o_apriori*, maybe *musica_wvp_apriori*, but how should I use them in a formula? Which variable in the data files represent the averaging kernels and how should I use them?

I think this technical information is important for data users. Could be explained in this article, or in a easily-accessible documentation?

**3  Minor comments**

- l 12: sensitivity -> sensitive

- l 14-15: grammar issue. Try removing "there is a ... data"

- l 30: constrained to -> interpreted in terms of the

- l 39: inevitable -> necessary

- l 43: missing citation

- l 44: missing , before Schneider

- l 45: TES did not end in 2012. It goes at least up to 2017, though with a reduced spatio-temporal sampling after a few years. See for example
  *https://tes.jpl.nasa.gov/data/plots/averages/monthly-mean-hdo-681hpa*

- l 45: you should also mention the AIRS data ([Worden et al., 2019]), which goes from 2002 to 2019 with a very good spatio-temporal sampling as well. See the AIRS $\delta D$ data publicly available on
  *https://avdc.gsfc.nasa.gov/pub/data/satellite/Aura/TES/.AIRs/TROPESS/HDO/*

- l 60: what is CF?

- l 109, 115: problem with this citation: why is the first name in the citation?

- Regarding data availability, I was able to download the data from
  *https://ncview.scc.kit.edu/thredds/musica_iasi/H2Oiso_v2/catalog.html* . However, at some point after following links I arrived on the page
  *https://radar.kit.edu/radar/en/dataset/dpYpVSBLOLrJZjxZ* and there, all I could download was land-masks: is it normal?

**References**

[Worden et al., 2019] Worden, J. R., Kulawik, S. S., Fu, D., Payne, V. H., Lipton, A. E., Polonsky, I., He, Y., Cady-Pereira, K., Moncet, J.-L., Herman, R. L., et al. (2019). Characterization and evaluation of airs-based estimates of the deuterium content of water vapor. *Atmospheric Measurement Techniques*, 12(4):2331–2339.

---

## Author Comment (AC1)

**Author Response to the Reviewer Comments to the manuscript "The MUSICA IASI {H₂O, δD} pair product" [essd-2021-87] submitted to Earth System and Science Data.**

We would kindly thank the Editor Nellie Elguindi for coordinating the Review Process as well as Camille Risi and one anonymous referee for their reviews. These have been very useful for improving the dataset as well as the manuscript. The individual comments are listed below (shown in red) including our responses (shown in black).

The changes discussed in this reply will be included in the revised manuscript and dataset and will thus become visible after re-submission.

Response to Reviewer 1:

**General comments**
"This paper presents a re-processed dataset of paired water vapor and δD retrievals from satellite observations with the IASI instrument. This is very useful dataset for the community of people working on water vapor isotopes. To date, this dataset will become the isotopic dataset with the highest spatio-temporal sampling (twice daily) at the global scale with the longer time period (six years). This submission is thus well-suited for publication in ESS.
The article is well-written and well-illustrated.
I expect that many users will be interested in using this data. However, the complexity of the dataset and of its use may be dissuasive. I propose a few suggestions in the specific comments.
Note that I'm just a data-user. I'm not expert in the satellite data processing, so I cannot comment on this aspect. I hope that at least another reviewer will be such an expert."

Thank you very much for this very constructive and positive feedback!

**Specific comments**

**"Optional: what about adding a user-friendly, fully gridded product?**
The full dataset is extremely heavy. IASI is not the first global dataset of water vapor isotopic composition by satellite, and my impression is that previous such datasets have been under-exploited. I can feel that many users would like to use such datasets, but the size of the data, and the complexity of its processing by users, is dissuasive for many people.
Therefore, I think it would be very useful for the community to build a simpler, lighter, regularly-gridded, user-friendly product. For example, it could be one netcdf file for each month, with a daily resolution, a 1x1 resolution, and with a coarse, horizontally-uniform vertical grid in altitude or pressure. It would contain quality filtered humidity and δD profiles and just a few ancillary information. Maybe you already made such a product, for example when plotting some gridded maps for this article?
Have you ever considered making such a product publicly available?"

Thank you for motivating the idea of creating an additional dataset more suitable for a wider range of end users. Until now, such a product has not been available (the gridded maps included in this article were only temporally derived from the comprehensive L2 products). However, we understand the purpose and usage of such a re-gridded dataset that focuses on the main products, i.e. pairs of $H_2O$ and δD for the free-troposphere.

Based on this comment, we decided to develop a more user-friendly MUSICA IASI L3 product of the {$H_2O$, δD} pairs and to publish it together with the here presented MUSICA IASI L2 product. For this purpose, we will regrid the quality-filtered MUSICA IASI L2 {$H_2O$, δD} pairs on a horizontal

1°x1° grid via averaging, interpolate the data on altitudes of main interest (2.95km, 4.22km and 6.38km) and reduce the output variable list to the main product variables (e.g. $H_2O$, $\delta D$, errors, temperature, pressure and some diagnostic metrics describing properties and uncertainties of the averaging). The L3 dataset will have an own DOI and will be derived and provided for the full time period presented in this manuscript.

Therefore, we will extend the MUSICA IASI processing scheme shown in Figure 1 with the "Re-gridding" step. Further, we will add a subsection 2.5 in Section 2 in this manuscript, where we will give a short summary on technical details of the new regridded MUSICA IASI L3 product. Additionally, we will extend Section 4.4 by including the documentation of the uncertainty that results from averaging the $H_2O$ and $\delta D$ values within the individual grid boxes.

**"Clarify how the data should be used**
**[...]** I think this technical information is important for data users. Could be explained in this article, or in a easily-accessible documentation?**"**

Thank you for this feedback. To create a technical user guide, which includes details about the output files and the included variables together with recommendations for how to work with the data, is a very good suggestion. We will provide such a documentation for both MUSICA IASI L2 and L3 datasets, and will add it as supplement to this manuscript. The more specific details given in the responses of the following questions will also be part of the documentation.

"Table 1: it is indicated that the flag eumetsat_cloud_summary_flag should be 1 or 2. However, l 337, the condition is written as eumetsat_cloud_summary_flag≤2. So should we select or discard values of eumetsat_cloud_summary_flag that are 0?"

The variable eumetsat_cloud_summary_flag does not contain any values equal to 0, therefore only data with values of 1 or 2 have to be considered. We will update Table 1, such that this information will be clearer. Additionally, we will document this in more detail in the MUSICA IASI {$H_2O$, $\delta D$} pair product user guide.

"I don't understand what is the musica_species_id dimension in the data files. For example, how do you plot figure 7 using the variable musica_wvp_dofs?"

In the output of the MUSICA IASI L2 {$H_2O$, $\delta D$} pair data, we also provide various retrieval matrices (averaging kernels, uncertainty matrices, ..) for the water vapour states. As described in the manuscript, the MUSICA IASI retrieval uses the state vectors { 0.5 * (log($H_2O$) +log(HDO)); log(HDO)-log($H_2O$) } as proxies for $H_2O$ and $\delta D$. For example, the DOFS is derived for both water vapour proxy states, therefore, following the convention of the MUSICA IASI files, the DOFS values of both proxy states are stored within a single netCDF variable. Thereby the two different water vapour proxy states are identified by means of the dimension `musica_species_id`. By doing so, the DOFS variable `musica_wvp_dofs` has the dimensions (`observation_id`, `musica_species_id`) and the DOFS values of the two water vapour proxy states can be read as follows:

```
**DOFS of first water vapour proxy state:**
dofs_wv1 = nc['musica_wvp_dofs'][:,0]
**DOFS of second water vapour proxy state:**
dofs_wv2 = nc['musica_wvp_dofs'][:,1]
```

However, as the MUSICA IASI {$H_2O$, $\delta D$} pair processing achieves that the two water vapour proxy states have practically the same averaging kernels, therefore also the DOFS values are quasi identical for both proxy states. For creating Figure 7, we plotted `musica_wvp_dofs` for

the second water vapour proxy state (second entry of `musica_species_id`), but the results would be overall similar when using the first water vapour proxy state (see Fig. R1).

[Figure]

**Figure R1** *Relative differences of the DOFS distributions of the first and second water vapour proxy states.*

"If I want to compare model outputs to the IASI data, is there a simple formula that would allow me to convolve model outputs with the IASI averaging kernels? I expect that I would have to use musica_h2o_apriori, maybe musica_wvp_apriori, but how should I use them in a formula? Which variable in the data files represent the averaging kernels and how should I use them?"

The appropriate formula would be

$$\hat{x} = A(x - x_a) + x_a$$

where $A$ is the averaging kernel matrix and $x_a$ the a priori profiles. As the averaging kernels are given in the water vapour proxy state base, the corresponding a priori profiles musica_wvp_apriori have to be used. Consequently, also the model profiles $x$ need to be transformed into the water vapour proxy state base, and the new adjusted model profile $\hat{x}$ could then be transformed back to $H_2O$ and $\delta D$.

As documented in the manuscript, the averaging kernels are stored in a decomposed way, thereby the variables musica_avk_wvp_lvec (matrix U), musica_avk_wvp_rvec (matrix V), musica_avk_wvp_val (matrix D) and musica_avk_wvp_rank (dimension of D) have to be used to reconstruct the kernel matrix $A$. We will include a technical description together with coding examples in the MUSICA IASI user guide to illustrate how this matrix reconstruction can be achieved.

**Minor comments**

"l 12: sensitivity -> sensitive"

Ok, this will be corrected.

"l 14-15: grammar issue. Try removing "there is a … data""

We will change this sentence to: "[...] but also higher latitudes show a considerable amount of reliable data".

"l 30: constrained to -> interpreted in terms of the"

Ok, we will rephrase this.

"l 39: inevitable -> necessary"

Ok, this will be rephrased.

"l 43: missing citation"

We will add the missing citation: Worden et al. (2012)

Worden, J., Kulawik, S., Frankenberg, C., Payne, V., Bowman, K., Cady-Peirara, K., Wecht, K., Lee, J. E., and Noone, D.: Profiles of CH4, HDO, H 2O, and N 2O with improved lower tropospheric vertical resolution from Aura TES radiances, Atmospheric Measurement Techniques, 5, 397–411, https://doi.org/10.5194/amt-5-397-2012, 2012.

"l 44: missing , before Schneider"

Ok, we will add the missing comma.

"l 45: TES did not end in 2012. It goes at least up to 2017, though with a reduced spatio-temporal sampling after a few years. See for example https://tes.jpl.nasa.gov/data/plots/averages/monthly-mean-hdo-681hpa"

Thank you for detecting this inaccurate wording.  Following Worden et al. (2019), in November 2009, an instrument degradation lead to a decrease in the spatio-temporal sampling. We checked the website
https://tes.jpl.nasa.gov/tes/data
and could find available TES L2 and L3 HDO data until 2012, and very few L3 maps for the end of 2017 and beginning of 2018.

We will adjust the corresponding sentence in our manuscript.

"l 45: you should also mention the AIRS data ([Worden et al., 2019]), which goes from 2002 to 2019 with a very good spatio-temporal sampling as well. See the AIRS δD data publicly available on https://avdc.gsfc.nasa.gov/pub/data/satellite/Aura/TES/.AIRs/TROPESS/HDO/"

Thank you for pointing out the increased data availability of the AIRS data, which we were not aware of, as it is not documented in Worden et al. (2019). However, we will add an according comment on the full data availability of AIRS.

"l 60: what is CF?"

CF stands for "Climate and Forecast" (see https://cfconventions.org/). The conventions for CF metadata define a standard for setting up Earth Science files. It includes standard names, definitions and types for variables and metadata, such that different climate and weather datasets can be created in a homogenized way.

We will add an according note to clarify the meaning of CF.

"l 109, 115: problem with this citation: why is the first name in the citation?"

Thanks for pointing this out. The problem with this citation is now solved.

"Regarding data availability, I was able to download the data from https://ncview.scc.kit.edu/thredds/musica_iasi/H2Oiso_v2/catalog.html . However, at some point after following links I arrived on the page

https://radar.kit.edu/radar/en/dataset/dpYpVSBLOLrJZjxZ and there, all I could download was land-masks: is it normal?"

This problem could be adressed and solved in personal communication with the referee. The download of the exemplary datasets via the page https://radar.kit.edu/radar/en/dataset/dpYpVSBLOLrJZjxZ should work now fine.

**Response to Reviewer 2:**

"This manuscript presents water vapor and its isotope (delta D) from the IASI satellite. The manuscript is well written, has a significant contribution to the community, and is worthy of prompt publication. I recommend it be accepted for publication, with a few minor revisions."

Thank you very much for this positive feedback!

"One of the main suggestions is whether provide a fully gridded product in different resolutions. For example, 0.25 degrees with 12-hour intervals or monthly. It would be great for users (like me). Currently, it is not friendly at all."

Thanks for this suggestion of creating a more user-friendly version of our MUSICA IASI dataset. Based on this comment and the similar comment of the first referee we decided to develop and derive a re-gridded L3 product of the comprehensive MUSICA IASI L2 {$H_2O$, $\delta D$} pair data. This dataset will comprise the full MUSICA IASI data time period, and we will add a further section in our manuscript, where we will give a coarse description of the new L3 dataset (see response to referee 1).

"Second, can the authors compare IASI products with other satellite products (such as TES or AIRS, if there is an overlap )? I think it is easy for us to understand the difference among the datasets (products)."

Thank you for bringing up this very valid point. We understand that the inter-comparison between $H_2O$ and $\delta D$ data from the different satellite platforms is an important task, in particular from scientific and technical point of views.  Such analyses would support the understanding of the datasets and their differences, but would also be crucial for creating synergies between the different datasets.

Currently, the MUSICA IASI data mainly overlap with data from AIRS and from TROPOMI. The comparison with the latter is already part of an ongoing project, see https://eo4society.esa.int/projects/sentinel-5p-innovation-water-vapour-isotopologues-project-h2o-iso/

Further inter-comparison of the MUSICA IASI data with AIRS and TROPOMI data will be part of the EUREC4A-iso campaign (https://eurec4a.eu/ ). Additionally, it is planned to extend the MUSICA IASI data period to the beginning of the operating missions of the Metop satellites (back to 2007). In that case, comprehensive inter-comparisons also with TES and SCIAMACHY would be possible.

However, as the different satellite products show typically very different characteristics, inter-comparisons between them should be carefully performed and interpreted, and should therefore be part of designated studies.

The intent of this publication is to serve as reference for our new MUSICA IASI datasets, which is why we have chosen this data journal (ESSD) for publication. Thus, we believe that the inter-comparison and interpretation of our MUSICA IASI data with other satellites exceeds the scope of this manuscript and this journal.

"L43: ? --> citation"

We will add the missing citation: Worden et al. (2012)

*Worden, J., Kulawik, S., Frankenberg, C., Payne, V., Bowman, K., Cady-Peirara, K., Wecht, K., Lee, J. E., and Noone, D.: Profiles of CH4, HDO, H2O, and N2O with improved lower tropospheric vertical resolution from Aura TES radiances, Atmospheric Measurement Techniques, 5, 397–411, https://doi.org/10.5194/amt-5-397-2012, 2012.*

"L44: ? --> wrong citation"

A ';' was missing between the explanation of the abbreviation and the citation, which is now corrected.

"L393: 1X1 degree?"

Throughout the whole document, we consistently use the notation 1°x1°, therefore we suggest to use it also in L393.

"Table 1: is it possible to add the information about flag values in the table? e.g., what is the meaning of the value of 1?"

We will extend Table 1 with the meaning of each flag value, such that this table is more intuitive to read. This information will also be part of the newly created technical user guide that will be provided in the supplement.

**Further changes:**

- During the review process of this manuscript, we were able to extend the period of available MUSICA IASI {$H_2O$, $\delta D$} pair data from June 2019 to December 2020. Accordingly, we will adjust the parts of the manuscript referring to the full data period. This includes to update the timeseries of $H_2O$ and $\delta D$ (Figure 12 of the initial manuscript).

- We decided to change Figure 11 from showing monthly averages of the total errors of $H_2O$ and $\delta D$ to showing the errors for a single day. In that way, it will be better comparable to the distributions of $H_2O$ and $\delta D$ in Figure 10, and it will show the real error values without considering any averaging.

---

## Author Response (AR1)

**Author Response to the Reviewer Comments to the manuscript "The MUSICA IASI {H$_2$O, δD} pair product" [essd-2021-87] submitted to Earth System and Science Data.**

We would kindly thank the Editor Nellie Elguindi for coordinating the Review Process as well as Camille Risi and one anonymous referee for their reviews. These have been very useful for improving the dataset as well as the manuscript. The individual comments are listed below (shown in red) including our responses (shown in black) and the changes made in the manuscript (shown in black and italics).

**Response to Reviewer 1:**

**General comments**

"This paper presents a re-processed dataset of paired water vapor and δD retrievals from satellite observations with the IASI instrument. This is very useful dataset for the community of people working on water vapor isotopes. To date, this dataset will become the isotopic dataset with the highest spatio-temporal sampling (twice daily) at the global scale with the longer time period (six years). This submission is thus well-suited for publication in ESS.
The article is well-written and well-illustrated.
I expect that many users will be interested in using this data. However, the complexity of the dataset and of its use may be dissuasive. I propose a few suggestions in the specific comments.
Note that I'm just a data-user. I'm not expert in the satellite data processing, so I cannot comment on this aspect. I hope that at least another reviewer will be such an expert."

Thank you very much for this very constructive and positive feedback!

**Specific comments**

**"Optional: what about adding a user-friendly, fully gridded product?**
The full dataset is extremely heavy. IASI is not the first global dataset of water vapor isotopic composition by satellite, and my impression is that previous such datasets have been under-exploited. I can feel that many users would like to use such datasets, but the size of the data, and the complexity of its processing by users, is dissuasive for many people.
Therefore, I think it would be very useful for the community to build a simpler, lighter, regularly-gridded, user-friendly product. For example, it could be one netcdf file for each month, with a daily resolution, a 1x1 resolution, and with a coarse, horizontally-uniform vertical grid in altitude or pressure. It would contain quality filtered humidity and δD profiles and just a few ancillary information. Maybe you already made such a product, for example when plotting some gridded maps for this article?
Have you ever considered making such a product publicly available?"

Thank you for motivating the idea of creating an additional dataset more suitable for a wider range of end users. Until now, such a product has not been available (the gridded maps included in this article were only temporally derived from the comprehensive Level-2 products). However, we understand the purpose and usage of such a re-gridded dataset that focuses on the main products, i.e. pairs of H$_2$O and δD for the free-troposphere.

Based on this comment, we decided to develop a more user-friendly MUSICA IASI Level-3 product of the {H$_2$O, δD} pairs and to publish it together with the here presented MUSICA IASI Level-2 product. For this purpose, we re-gridded the quality-filtered MUSICA IASI L2 {H$_2$O, δD} pairs on a horizontal 1°x1° grid via averaging, interpolated the data on altitudes of main interest

(2.95km, 4.22km and 6.38km) and reduced the output variable list to the main product variables (e.g. $H_2O$, $\delta D$, errors, temperature, pressure and some diagnostic metrics describing properties and uncertainties of the averaging). The Level-3 dataset has an own DOI (https://doi.org/10.5194/essd-2021-87) and comprises the full time period presented in this manuscript.

Therefore, we extended the MUSICA IASI processing scheme shown in Figure 1 with the "Re-gridding" step. Further, we added the section "2.5 Generation of a re-gridded {H2O, $\delta D$} pair product" in this manuscript, where we give a short summary on technical details of the new regridded MUSICA IASI Level-3 product:

*"2.5 Generation of a re-gridded {$H_2O$, $\delta D$} pair product*

*For reasons of traceability and data re-usage, the output files produced by the MUSICA IASI {$H_2O$, $\delta D$} pair post-processing include arrays for reconstructing different retrieval metrics, such as the averaging kernels and uncertainty covariances. Consequently, the corresponding files have high computational requirements with respect to storing and processing. Therefore, we generate an additional Level-3 dataset, the purpose of which is to allow for a simplified and less computationally intensive application of the MUSICA IASI {$H_2O$, $\delta D$} pairs.*

*As the MUSICA {$H_2O$, $\delta D$} pair product has highest sensitivity typically in the mid-troposphere (between 2.9 and 6.4 km a.s.l.), the Level-3 dataset comprises all quality filtered {$H_2O$, $\delta D$} pairs (according to Table 1) for the fixed altitude levels at 2.9, 4.2 and 6.4 km (analogous to Fig. 2) and re-gridded on a regular 1°x1° grid. The latter is achieved by linear averaging all data of $H_2O$ and HDO (derived from $\delta D$) within the individual grid boxes and the a posteriori calculation of an averaged $\delta D$ based on the averaged $H_2O$ and HDO data (according to Eqn. (1)). In case of the total errors of $H_2O$ and $\delta D$, the averaging of their distributions on the regular grid requires to take into account the nature of the errors, i.e. the relative contributions of systematic and random error components. This is crucial, because averaging over systematic error components will balance around a constant systematic bias, whereas the random errors will get smaller the more data points are used for averaging. Here, we follow the simple assumption that errors due to measurement noise and temperature consist of 50% systematic and 50% random error components. We accordingly convolute all measurement noise and temperature error values within the individual grid boxes and afterwards form the total $H_2O$ and $\delta D$ errors for each grid box. Furthermore, we provide a metric indicating the representativeness of the averaged $H_2O$ and $\delta D$. It is the RMS of the differences of the individual $H_2O$ and $\delta D$ values within the grid boxes to their respective averages. For $H_2O$, the respective calculations are made on the logarithmic scale. As this metric is a measure for how scattered the individual data are within a single grid box, it indicates the data range for which the averaged value of a single grid box is representative."*

Additionally, we adjusted Section 3 by documenting the data availability of the new Level-3 dataset:

*"The Level-3 product described in Sect. 2.5 is generated for all files of the Level-2 dataset, with an approximate individual file size of 4 MB and full output volume of 22 GB. The metadata of the Level-2 and Level-3 output NetCDF4 files are in agreement with the CF metadata conventions."*

Section 4.4 is modified by adding a new Figure (Figure 12 in revised manuscript) which shows the uncertainty that results from averaging the $H_2O$ and $\delta D$ values within the individual grid boxes. Following paragraph is accordingly added in the text:

"*Furthermore, Fig. 12 provides an overview of the representativeness of the $H_2O$ and $\delta D$ data being averaged for the regular 1°×1° grid, as done for the Level-3 dataset of the MUSICA IASI {$H_2O$, $\delta D$} pairs (see Sect. 2.5). For this purpose, monthly means of the representativeness metrics (RMS values, see Sect. 2.5) are shown for Februar and August 2018. The lowest RMS values appear for both $H_2O$ and $\delta D$ over oceans, meaning that these regions exhibit rather homogeneous and compact distributions in $H_2O$ and $\delta D$ within the individual grid boxes. In contrast, the highest RMS values arise for coastal regions in the subtropics, where due to the land-sea-contrast the largest spread of $H_2O$ and $\delta D$ values within the individual grid boxes develops.*"

**"Clarify how the data should be used**
**[...]** I think this technical information is important for data users. Could be explained in this article, or in a easily-accessible documentation?**"**

Thank you for this feedback. To create a technical user guide, which includes details about the output files and the included variables together with recommendations for how to work with the data, is a very good suggestion. We put together such a documentation for both MUSICA IASI L2 and L3 datasets, and added it as supplement to this manuscript. The more specific details given in the responses of the following questions are also part of the documentation.

"Table 1: it is indicated that the flag eumetsat_cloud_summary_flag should be 1 or 2. However, l 337, the condition is written as eumetsat_cloud_summary_flag≤2. So should we select or discard values of eumetsat_cloud_summary_flag that are 0?"

The variable eumetsat_cloud_summary_flag does not contain any values equal to 0, therefore only data with values of 1 or 2 have to be considered. We updated Table 1, such that this information becomes clearer. Additionally, we documented this in more detail in the MUSICA IASI {$H_2O$, $\delta D$} pair product user guide.

"I don't understand what is the musica_species_id dimension in the data files. For example, how do you plot figure 7 using the variable musica_wvp_dofs?"

In the output of the MUSICA IASI L2 {$H_2O$, $\delta D$} pair data, we also provide various retrieval matrices (averaging kernels, uncertainty matrices, ..) for the water vapour states. As described in the manuscript, the MUSICA IASI retrieval uses the state vectors { 0.5 * (log($H_2O$) +log(HDO)); log(HDO)-log($H_2O$) } as proxies for $H_2O$ and $\delta D$. For example, the DOFS is derived for both water vapour proxy states, therefore, following the convention of the MUSICA IASI files, the DOFS values of both proxy states are stored within a single netCDF variable. Thereby the two different water vapour proxy states are identified by means of the dimension `musica_species_id`. By doing so, the DOFS variable `musica_wvp_dofs` has the dimensions (`observation_id`, `musica_species_id`) and the DOFS values of the two water vapour proxy states can be read as follows:

```
**DOFS of first water vapour proxy state:**
dofs_wv1 = nc['musica_wvp_dofs'][:,0]
**DOFS of second water vapour proxy state:**
dofs_wv2 = nc['musica_wvp_dofs'][:,1]
```

However, as the MUSICA IASI {$H_2O$, $\delta D$} pair processing achieves that the two water vapour proxy states have practically the same averaging kernels, therefore also the DOFS values are quasi identical for both proxy states. For creating Figure 7, we plotted `musica_wvp_dofs` for the second water vapour proxy state (second entry of `musica_species_id`), but the results would be overall similar when using the first water vapour proxy state (see Fig. R1).

[Figure]

**Figure R1** *Relative differences of the DOFS distributions of the first and second water vapour proxy states.*

*"If I want to compare model outputs to the IASI data, is there a simple formula that would allow me to convolve model outputs with the IASI averaging kernels? I expect that I would have to use musica_h2o_apriori, maybe musica_wvp_apriori, but how should I use them in a formula? Which variable in the data files represent the averaging kernels and how should I use them?"*

The appropriate formula would be

$$\hat{x} = A(x - x_a) + x_a$$

where $A$ is the averaging kernel matrix and $x_a$ the a priori profiles. As the averaging kernels are given in the water vapour proxy state base, the corresponding a priori profiles musica_wvp_apriori have to be used. Consequently, also the model profiles $x$ need to be transformed into the water vapour proxy state base, and the new adjusted model profile $\hat{x}$ could then be transformed back to $H_2O$ and $\delta D$.

As documented in the manuscript, the averaging kernels are stored in a decomposed way, thereby the variables musica_avk_wvp_lvec (matrix U), musica_avk_wvp_rvec (matrix V), musica_avk_wvp_val (matrix D) and musica_avk_wvp_rank (dimension of D) have to be used to reconstruct the kernel matrix $A$. We included a technical description together with coding examples in the MUSICA IASI user guide to illustrate how this matrix reconstruction can be achieved.

**Minor comments**

*"l 12: sensitivity -> sensitive"*

Ok, this is corrected.

*"l 14-15: grammar issue. Try removing "there is a ... data""*

We changed this sentence to:

*"[...] but also higher latitudes show a considerable amount of reliable data".*

*"l 30: constrained to -> interpreted in terms of the"*

Ok, we rephrased this.

"l 39: inevitable -> necessary"

Ok, this is rephrased.

"l 43: missing citation"

We added the missing citation: Worden et al. (2012)

Worden, J., Kulawik, S., Frankenberg, C., Payne, V., Bowman, K., Cady-Peirara, K., Wecht, K., Lee, J. E., and Noone, D.: Profiles of CH4, HDO, H 2O, and N 2O with improved lower tropospheric vertical resolution from Aura TES radiances, Atmospheric Measurement Techniques, 5, 397–411, https://doi.org/10.5194/amt-5-397-2012, 2012.

"l 44: missing , before Schneider"

Ok, we added the missing comma.

"l 45: TES did not end in 2012. It goes at least up to 2017, though with a reduced spatio-temporal sampling after a few years. See for example https://tes.jpl.nasa.gov/data/plots/averages/monthly-mean-hdo-681hpa"

Thank you for detecting this inaccurate wording.  Following Worden et al. (2019), in November 2009, an instrument degradation lead to a decrease in the spatio-temporal sampling. We checked the website
https://tes.jpl.nasa.gov/tes/data
and could find available TES L2 and L3 HDO data until 2012, and very few Level-3 maps for the end of 2017 and beginning of 2018.

We adjusted the corresponding sentence in our manuscript as follows:

"*Up to now, only few global and long-time referenced space-borne datasets of tropospheric HDO/H2O are available, e.g. total column data from the sensor SCIAMACHY (Scanning Imaging Absorption Spectrometer for Atmospheric Chartography; Schneider et al., 2018) between 2003 to 2012, profile data from TES (Tropospheric Emission Spectrometer; Worden et al., 2012) between 2004 to 2012 (with few available data between 2017 and 2018) and from AIRS (Atmospheric Infrared Sounder; Worden et al., 2019) between 2002 to early 2020. The maximum data availability of these datasets ranges in the order of 1000 to 30,000 observations per day.*"

"l 45: you should also mention the AIRS data ([Worden et al., 2019]), which goes from 2002 to 2019 with a very good spatio-temporal sampling as well. See the AIRS δD data publicly available on https://avdc.gsfc.nasa.gov/pub/data/satellite/Aura/TES/.AIRs/TROPESS/HDO/"

Thank you for pointing out the increased data availability of the AIRS data, which we were not aware of, as it is not documented in Worden et al. (2019). However, we added an according comment on the full data availability of AIRS (see comment above).

"l 60: what is CF?"

CF stands for "Climate and Forecast" (see https://cfconventions.org/). The conventions for CF metadata define a standard for setting up Earth Science files. It includes standard names, definitions and types for variables and metadata, such that different climate and weather datasets can be created in a homogenized way.

We added an according note to clarify the meaning of CF:

"*The chosen output format is NetCDF4 and the metadata are in agreement with the conventions for CF (Climate and Forecast) metadata (version 1.7, see https://cfconventions.org/)*"

"l 109, 115: problem with this citation: why is the first name in the citation?"

Thanks for pointing this out. The problem with this citation is now solved.

"Regarding data availability, I was able to download the data from https://ncview.scc.kit.edu/thredds/musica_iasi/H2Oiso_v2/catalog.html . However, at some point after following links I arrived on the page https://radar.kit.edu/radar/en/dataset/dpYpVSBLOLrJZjxZ and there, all I could download was land-masks: is it normal?"

This problem could be adressed and solved in personal communication with the referee. The download of the exemplary datasets via the page https://radar.kit.edu/radar/en/dataset/dpYpVSBLOLrJZjxZ should work now fine.

**Response to Reviewer 2:**

"This manuscript presents water vapor and its isotope (delta D) from the IASI satellite. The manuscript is well written, has a significant contribution to the community, and is worthy of prompt publication. I recommend it be accepted for publication, with a few minor revisions."

Thank you very much for this positive feedback!

"One of the main suggestions is whether provide a fully gridded product in different resolutions. For example, 0.25 degrees with 12-hour intervals or monthly. It would be great for users (like me). Currently, it is not friendly at all."

Thanks for this suggestion of creating a more user-friendly version of our MUSICA IASI dataset. Based on this comment and the similar comment of the first referee we decided to develop and derive a re-gridded L3 product of the comprehensive MUSICA IASI L2 {$H_2O$, $\delta D$} pair data. This dataset comprises the full MUSICA IASI data time period, and we added a further section in our manuscript, where we give a coarse description of the new L3 dataset (see response to referee 1).

"Second, can the authors compare IASI products with other satellite products (such as TES or AIRS, if there is an overlap )? I think it is easy for us to understand the difference among the datasets (products)."

Thank you for bringing up this very valid point. We understand that the inter-comparison between $H_2O$ and $\delta D$ data from the different satellite platforms is an important task, in particular from scientific and technical point of views. Such analyses would support the understanding of the datasets and their differences, but would also be crucial for creating synergies between the different datasets.

Currently, the MUSICA IASI data mainly overlap with data from AIRS and from TROPOMI. The comparison with the latter is already part of an ongoing project, see https://eo4society.esa.int/projects/sentinel-5p-innovation-water-vapour-isotopologues-project-h2o-iso/
Further inter-comparison of the MUSICA IASI data with AIRS and TROPOMI data will be part of the EUREC4A-iso campaign (https://eurec4a.eu/ ). Additionally, it is planned to extend the MUSICA IASI data period to the beginning of the operating missions of the Metop satellites (back to 2007). In that case, comprehensive inter-comparisons also with TES and SCIAMACHY would be possible.

However, as the different satellite products show typically very different characteristics, inter-comparisons between them should be carefully performed and interpreted, and should therefore be part of designated studies.

The intent of this publication is to serve as reference for our new MUSICA IASI datasets, which is why we have chosen this data journal (ESSD) for publication. Thus, we believe that the inter-comparison and interpretation of our MUSICA IASI data with other satellites exceeds the scope of this manuscript and this journal.

"L43: ? --> citation"

We added the missing citation: Worden et al. (2012)

*Worden, J., Kulawik, S., Frankenberg, C., Payne, V., Bowman, K., Cady-Peirara, K., Wecht, K., Lee, J. E., and Noone, D.: Profiles of CH4, HDO, H2O, and N2O with improved lower tropospheric vertical resolution from Aura TES radiances, Atmospheric Measurement Techniques, 5, 397–411, https://doi.org/10.5194/amt-5-397-2012, 2012.*

"L44: ? --> wrong citation"

A ';' was missing between the explanation of the abbreviation and the citation, which is now corrected.

"L393: 1X1 degree?"

Throughout the whole document, we consistently use the notation 1°x1°, therefore we suggest to use it also in L393.

"Table 1: is it possible to add the information about flag values in the table? e.g., what is the meaning of the value of 1?"

We extended Table 1 with the meaning of each flag value, such that this table is more intuitive to read. This information is also part of the newly created technical user guide that is provided in the supplement.

**Further changes:**

- To underline the availability and documentation of the different datasets for the MUSICA IASI {$H_2O$, $\delta D$} pairs, we have changed the title of the manuscript to:
  *"The global and multi-annual MUSICA IASI {$H_2O, \delta D$} pair dataset"*

- During the review process of this manuscript, we were able to extend the period of available MUSICA IASI {$H_2O$, $\delta D$} pair data from June 2019 to December 2020. Accordingly, we adjusted the parts of the manuscript referring to the full data period. This includes to update the timeseries of $H_2O$ and $\delta D$ (Figure 12 of the initial manuscript).

- We decided to change Figure 11 from showing monthly averages of the total errors of $H_2O$ and $\delta D$ to showing the errors for a single day. In that way, it will be better comparable to the distributions of $H_2O$ and $\delta D$ in Figure 10, and it will show the real error values without considering any averaging.

- In Eqn. (22), the filter condition was erroneously defined with the threshold value 3.5, while actually it should be 4. Therefore, this is changed in the revised manuscript. As the figures of the initial manuscript were created by considering the threshold value 4, no further changes were necessary.